# Ito Diffusion Approximation of Universal Ito Chains for Sampling, Optimization and Boosting

**Aleksei Ustimenko**
ShareChat
aleksei.ustimenko@sharechat.co

**Aleksandr Beznosikov**
Innopolis University, Skoltech, RANEPA, Yandex
anbeznosikov@gmail.com

## Abstract

In this work, we consider rather general and broad class of Markov chains, Ito chains, that look like Euler-Maryama discretization of some Stochastic Differential Equation. The chain we study is a unified framework for theoretical analysis. It comes with almost arbitrary isotropic and state-dependent noise instead of normal and state-independent one as in most related papers. Moreover, in our chain the drift and diffusion coefficient can be inexact in order to cover wide range of applications as Stochastic Gradient Langevin Dynamics, sampling, Stochastic Gradient Descent or Stochastic Gradient Boosting. We prove the bound in $\mathcal{W}_2$-distance between the laws of our Ito chain and corresponding differential equation. These results improve or cover most of the known estimates. And for some particular cases, our analysis is the first.

## 1 Introduction

The connection between diffusion processes and homogeneous Markov chains has been investigated for a long time (Skorokhod, 1963). If we need to approximate the given diffusion by some homogeneous Markov chain, it is easy to realize because we are free to construct the chain nicely, meaning that we can choose terms and properties of MC, e.g., as it was shown in (Raginsky et al., 2017). However, often the inverse problem arises, namely, we have the a priori given chain, and the goal is to study it via the corresponding diffusion approximation. This task is an increasingly popular and hot research topic. Indeed, it is used to investigate different sampling techniques (Orvieto & Lucchi, 2018), to describe the behavior of optimization methods (Raginsky et al., 2017) and to understand the convergence of boosting algorithms (Ustimenko & Prokhorenkova, 2021). From practical experience, the given Markov chain may not have good properties that are easy to analyze in theory. Thus, the aim of our work is to study when diffusion approximation holds for as broad as the possible class of homogeneous Markov chains, i.e., we want to consider the maximally general chain and place the broadest possible assumptions on it whilst obtaining diffusion approximation guarantee.

The key and most popular MC is Langevin-based (Raginsky et al., 2017; Dalalyan, 2017; Cheng et al., 2018; Erdogdu et al., 2018; Durmus & Moulines, 2019; Orvieto & Lucchi, 2018; Cheng et al., 2020) (which corresponds to Langevin diffusion). Such a chain is found in most existing works. In this paper, we propose a more general Ito chain:

$$X_{k+1} = X_k + \eta\big(b(X_k) + \delta_k\big) + \sqrt{\eta^{1+\gamma}}\big(\sigma(X_k) + \Delta_k\big)\epsilon_k(X_k), \tag{1}$$

where $X_k \in \mathbb{R}^d$ is the state of the chain at the moment $k \geq 0$, $\eta \in \mathbb{R}$ is the stepsize, $b : \mathbb{R}^d \to \mathbb{R}^d$ is the main part in the drift of the chain (e.g., for Langevin MC, $b = -\nabla f$, where $f$ is some potential), $\delta \in \mathbb{R}^d$ is the deterministic bias of the drift (Dalalyan & Karagulyan, 2019) (e.g., such bias occurs if we use smoothing techniques (Chatterji et al., 2020) for Langevin dynamics, or a gradient-free method instead of a gradient based in optimization (Duchi et al., 2015), or a special modification of boosting (Ustimenko & Prokhorenkova, 2021)), parameter $\gamma$ takes values between 0 and 1 (e.g., for SGD case $\gamma = 1$, for sampling case $\gamma = 0$), the whole expression $\sqrt{\eta^{1+\gamma}}(\sigma + \Delta)\epsilon$ is responsible for generally non-Gaussian noise, depending on the current state of our chain. In this case, $\sigma : \mathbb{R}^d \to \mathbb{R}^{d \times d}$ is called the covariance coefficient, $\Delta \in \mathbb{R}^{d \times d}$ – the covariance shift and $\epsilon : \mathbb{R}^d \to \mathbb{R}^d$ – the noise function.

Next, let us consider the following Ito diffusion:

$$\mathrm{d}Z_t = b(Z_t)\mathrm{d}t + \sqrt{\eta^\gamma}\sigma(Z_t)\mathrm{d}W_t. \tag{2}$$

with Brownian motion $W_t$. This is the diffusion for which we obtain bounds on the discretization error with equation 1.

## 1.1 OUR CONTRIBUTION AND RELATED WORKS

Explanation of our contribution can be divided into three parts: 1) the universality of the chain equation 1, 2) rather weak and broad assumptions on the chain's terms, 3) guarantees on the discretization error between equation 1 and equation 2 in the $\mathcal{W}_2$-distance.

• **Unified framework.** The Ito chain equation 1 incorporates a variety of practical approaches and techniques – see Table 1. In particular, equation 1 can be used to describe:

*Dynamics.* Primarily, chain equation 1 is suitable for analyzing Langevin Dynamics, which have a wide range of applications. Here we can note the classical results in sampling (Ma et al., 2019; Chatterji et al., 2020; Dalalyan, 2017; Durmus et al., 2019; Durmus & Moulines, 2019), continuous optimization (Gelfand et al., 1992), as well as modern and hot techniques in generative models (Gidel et al., 2018).

*SGD and beyond.* The use of equation 1 is also a rather popular way to study the behavior of stochastic optimization methods. In particular, one can highlight papers on SGD (Robbins & Monro, 1951; Ankirchner & Perko, 2021; Orvieto & Lucchi, 2018; Hu et al., 2017) analysis via diffusions, as well as works (Bhardwaj, 2019; Kim et al., 2020) about diffusions for popular and famous modifications of the original SGD: SGD with momentum (Robbins & Monro, 1951), RMSProp (Tieleman et al., 2012) and Adam (Kingma & Ba, 2014). Moreover, equation 1 can be used to describe stochastic methods not only for minimization but for saddle point problems (Facchinei & Pang, 2003; Ben-Tal et al., 2009; Juditsky et al., 2011; Gidel et al., 2018): $\min_X \max_Y f(X,Y)$, and fixed point problems (Bailion et al., 1978): $F(X^*) = X^*$.

*Gradient Boosting.* Moreover, Gradient Boosting algorithms, in particular, the original one from (Friedman, 2001), and the Langevin-based boosting proposed in (Ustimenko & Prokhorenkova, 2021), can be written in the form equation 1.

Table 1: Matching different methods and frameworks with the parameters of the Ito chain equation 1 and Assumption 1.

| | Case | $\gamma$ | $\alpha$ | $\beta$ | $b(X_k)$ | $\delta_k$ | $\sigma(X_k)$ | $\Delta_k$ |
|---|---|---|---|---|---|---|---|---|
| **Dynamics** | GLD | 0 | $\infty$ [1] | $\infty$ | $-\nabla_x f(X_k)$ | 0 | $\sqrt{\frac{2}{\tau}}I_d$ [2] | 0 |
| | SGLD (Gelfand et al., 1992) | 0 | $\frac{1}{2}$ | 1 | $-\nabla_x f(X_k)$ | 0 | $\sqrt{\frac{2}{\tau}}I_d$ | $\sqrt{\frac{2}{\tau}I_d + \eta\mathrm{Cov}(\widehat\nabla)} - \sqrt{\frac{2}{\tau}}I_d$ |
| | SGLD with smoothing (Chatterji et al., 2020) | 0 | $\frac{1}{2}$ | 1 | $-\nabla_x f(X_k)$ | $\nabla_x\left(f(X_k) - \mathbb{E}_\varepsilon f(X_k + \eta^{\frac{1}{2}}\varepsilon)\right)$ | $\sqrt{\frac{2}{\tau}}I_d$ | $\sqrt{\frac{2}{\tau}I_d + \eta\mathrm{Cov}(\widehat\nabla)} - \sqrt{\frac{2}{\tau}}I_d$ |
| **Optimization** | SGD (Robbins & Monro, 1951) | 1 | $\infty$ | 0 | $-\nabla_x f(X_k)$ | 0 | $\sqrt{\mathrm{Cov}(\overline\nabla)}$ | 0 |
| | SGDA (Dem'yanov & Pevnyi, 1972) | 1 | $\infty$ | 0 | $\left(-\nabla_x f(X_k, Y_k), \nabla_\mathbf{y} f(X_k, Y_k)\right)$ | 0 | $\sqrt{\mathrm{Cov}(\overline\nabla)}$ | 0 |
| | SA-FP (Bailion et al., 1978) | 1 | $\infty$ | 0 | $F(X_k) - X_k$ | 0 | $\sqrt{\mathrm{Cov}(\overline F)}$ | 0 |
| | SA (Bailion et al., 1978) | 1 | $\infty$ | 0 | $H(X_k) - a$ [3] | 0 | $\sqrt{\mathrm{Cov}(\overline H)}$ | 0 |
| **Boosting** | SGB (Friedman, 2001) | 1 | $\infty$ | 0 | $-P(X_k)\nabla_x f(X_k)$ | 0 | $\sqrt{\mathrm{Cov}(\overline\nabla)}$ | 0 |
| | SGLB (Ustimenko & Prokhorenkova, 2021) [4] | 0 | $\frac{1}{2}$ | 0 | $-P(X_k)\nabla_x f(X_k)$ | 0 | $\sqrt{\frac{2}{\tau}}I_d$ | $\sqrt{\eta\mathrm{Cov}(\widehat\nabla)}$ |
| | SGLB-O (Ustimenko & Prokhorenkova, 2021) [5] | 0 | $\frac{1}{4}$ | 0 | $-P_\infty\nabla_x f(X_k)$ | $(P_\infty - P(X_k))\nabla_x f(X_k)$ | $\sqrt{\frac{2}{\tau}}I_d$ | $\sqrt{\eta\mathrm{Cov}(\widehat\nabla)}$ |

[1] $\eta^\infty$ means that the terms multiplied by it vanish, i.e., we can take $\alpha$ as large as we desire when calculating overall approximation error.

[2] $\tau$ refers to inverse diffusion temperature.

[3] $a$ is any constant. Stochastic Approximation tries to solve $H(x) = a$.

[4] SGLB here is defined as in the original paper, but here we ignore smoothing applied to the trees selection algorithm.

[5] "O" stands for "original", i.e., as presented in the original paper. In that case, such coefficients appear if we take the distribution of trees as in the paper.

• **Wide assumptions and results.** We consider the most general and practical setting for equation 1 - see Table 2. Next, we give more details on each of the columns of Table 2 (comparison criteria):

*Non-normality of noise.* The central and widely used assumption about noise in analyses of MC satisfying equation 1 (e.g., Langevin-based) is that it has a normal distribution (Raginsky et al., 2017; Dalalyan, 2017; Cheng et al., 2018; Durmus & Moulines, 2019; Ma et al., 2019; Feng et al., 2019; Orvieto & Lucchi, 2018; Chatterji et al., 2020; Xie et al., 2021). However, practice suggests otherwise. For example, stochastic gradient noise in the training of neural networks is not Gaussian for classical and small models (Simsekli et al., 2019), as well as for modern and large transformers (Zhang et al., 2020). Therefore, as in some papers on the SDE approximations for SGD (Li et al., 2019a;b; Hu et al., 2017; Cheng et al., 2020), we assume that the noise in our Ito chain equation 1 is non-Gaussian – see Assumption 1.

*Dependence of the noise on the current state.* Most of the papers on Langevin MCMC assume that the noise is independent of the current state (Raginsky et al., 2017; Erdogdu et al., 2018; Dalalyan, 2017; Cheng et al., 2018; Durmus & Moulines, 2019; Ma et al., 2019; Chatterji et al., 2020; Feng et al., 2019; Orvieto & Lucchi, 2018; Xie et al., 2021). However, let's talk primarily about SGD analysis. This assumption is often unmet because the noise in a stochastic gradient can depend strongly on the current weights of the model, e.g., how close we are to the optimal weights. Therefore, we consider the state-dependent noise in our chain equation 1. However, in our chain, we require that the diffusion coefficient is strictly non-singular and its minimal eigenvalue is lower bounded uniformly from zero (uniformly elliptic, see (Baldi & Baldi, 2017), p. 308), which is a limitation of our work compared to the analysis done in (Li et al., 2019a).

Table 2: Comparison of the theoretical setups and results on Markov chains and diffusions analysis.

| Reference | Noise | | Generator, i.e. $b(\cdot)$ | | Non-uniformly elliptic | $\mathcal{W}_2$ |
|---|---|---|---|---|---|---|
| | Distribution | Dependence | Non-convex | Non-dissipative | | |
| (Raginsky et al., 2017) | $\mathcal{N}$+SG | ✓ | ✓ | ✗ | ✓ | ✓ |
| (Dalalyan, 2017) | $\mathcal{N}$ | ✗ | ✗ | ✗ | ✗ | ✓ |
| (Cheng et al., 2018) | $\mathcal{N}$ | ✗ | ✓ | ✗ | ✗ | ✗ |
| (Erdogdu et al., 2018) | $\mathcal{N}$+SG | ✓ | ✓ | ✗ | ✗ | ✗ |
| (Durmus & Moulines, 2019) | $\mathcal{N}$ | ✗ | ✗ | ✗ | ✗ | ✓ |
| (Ma et al., 2019) | $\mathcal{N}$ | ✗ | ✓ | ✗ | ✗ | ✗ |
| (Li et al., 2019b) | $\mathcal{N}$ | ✓ | ✗ | ✗ | ✓ | ✓ |
| (Chatterji et al., 2020) | $\mathcal{N}$ | ✗ | ✗ | ✗ | ✗ | ✓ |
| (Feng et al., 2019) | $\|\epsilon\| \leq$ const a.s. | ✗ | ✓ | ✗ | ✗ | ✗ |
| (Orvieto & Lucchi, 2018) | $\mathcal{N}$ | ✗ | ✓ | ✗ | ✗ | ✗ |
| (Ankirchner & Perko, 2021) | $\mathcal{N}$ | ✗ | ✓ | ✗ | ✓ | ✗ |
| (Hu et al., 2017) | $\mathcal{N}$ | ✗ | ✓ | ✗ | ✗ | ✗ |
| (Xie et al., 2021) | $\mathcal{N}$ | ✗ | ✗ | ✗ | ✗ | ✗ |
| (Ustimenko & Prokhorenkova, 2021) | Mixture $\mathcal{N}$ | ✓ | ✓ | ✗ | ✓ | ✗ |
| (Cheng et al., 2020) | $\mathcal{N}$+SG | ✓ | ✓ | ✗ | ✗ | ✗ |
| (Li et al., 2019a) | $\mathbb{E}\|\epsilon\|^4 \leq$ const | ✓ | ✓ | ✗ | ✓ | ✗ |
| Ours | $\mathbb{E}\|\epsilon\|^4 \leq$ const | ✓ | ✓ | ✓ | ✗ | ✓ |

*Without convexity and dissipativity assumptions.* Note also that often, when dealing with Langevin MC, the authors consider the convex/monotone setup (Dalalyan, 2017; Erdogdu et al., 2018; Durmus & Moulines, 2019; Li et al., 2019b; Chatterji et al., 2020; Xie et al., 2021), which is possible and relevant, but at the same time restricted. This is primarily because a large number of practical problems (including ML problems) are non-convex: neural networks (Goodfellow et al., 2016), adversarial training (Goodfellow et al., 2014), games (Hazan et al., 2017), problems with specific losses (Nguyen & Sanner, 2013) and many others examples. However, even those works (Raginsky et al., 2017; Cheng et al., 2018; Ma et al., 2019; Feng et al., 2019; Orvieto & Lucchi, 2018; Ankirchner & Perko, 2021; Hu et al., 2017; Ustimenko & Prokhorenkova, 2021; Cheng et al., 2020) which consider the non-convex case make it under the dissipativity assumption (see for example, A.3 from (Cheng et al., 2020)). This assumption means non-convexity inside some ball and strong convexity outside the ball. However, it is not always fulfilled for practical problems and is primarily needed to simplify the analysis.

Here, in addition to examples of non-convex ML problems, we can mention already classic examples of the use of stochastic processes in applied problems (Øksendal & Øksendal, 2003). Even the simplest processes of these are not dissipative. For example, one of the best-known stochastic processes, the Ornstein–Uhlenbeck process, can be introduced using the diffusion equation 2: $dZ_t = \alpha Z_t dt + \sigma dW_t$, wit $\alpha, \sigma > 0$. This process has been known for almost a century (Uhlenbeck & Ornstein, 1930), and has both classical applications and completely new ones, e.g., in machine learning (Lillicrap et al., 2015; Blanc et al., 2020). In our analysis, we do not need assumptions about either convexity or dissipativity (formally, for our chain equation 1 we should talk not about non-convexity/non-

dissipativity, but about non-monotonicity of $(-b)$ because, unlike the Langevin MC, we do not assume that $b = -\nabla f$) – see Assumption 1.

$\mathcal{W}_2$ *convergence.* In our work, we give a bound on the discretization error in the $\mathcal{W}_2$-distance. This is a common criterion in many works (Raginsky et al., 2017; Dalalyan, 2017; Erdogdu et al., 2018; Li et al., 2019b; Ma et al., 2019). However, in the meantime, the main competitive paper (Cheng et al., 2020) gives weaker guarantees, namely, the distance is measured in $\mathcal{W}_1$. First of all, note that from the $\mathcal{W}_2$ bound follows the $\mathcal{W}_1$. The converse is not true. Thus $\mathcal{W}_2$-distance can be used when $b$ in equation 1 is a gradient of some potential/loss function $f$, and this potential is quadratic growth (e.g., MSE) – see Lemma 6 from (Raginsky et al., 2017). In turn, $\mathcal{W}_1$ cannot be used for such potentials. (Edwards, 2011)

• **Best and new rates.** We provide explicit bound between the laws of $X_k$ and $Z_{k\eta}$ in $\mathcal{W}_2$-distance for almost arbitrary noise:

$$\mathcal{W}_2\big(\mathcal{L}(X_k), \mathcal{L}(Z_{k\eta})\big) = \mathcal{O}\Big(\eta^\theta + \eta^{\frac{\theta}{2} + \frac{\gamma}{4}}\Big), \text{ where } \theta = \min\left\{\alpha; \frac{(\gamma+1)(1+\chi_0) + (\gamma+\beta)(1-\chi_0)}{4}\right\}$$

with $\chi_0 = \mathbb{1}\{\text{Gaussian noise}\}$. $\alpha$ and $\beta$ are parameters from Assumption 1, $\gamma$ is from the chain equation 1. Specific values for $\alpha$, $\beta$ and $\gamma$ in different cases can be found in Table 1. Using this table, it is possible to obtain convergence guarantees for the major cases – see Table 3.

For SGLD, our bounds are the best in the existing literature. In particular, they coincide with the results for the general Euler-Maryama discretization with normal noise – a subset of chains from the family equation 1. For non-Gaussian noise, our results are first in the literature.

SGD with non-Gaussian noise was considered in (Cheng et al., 2020), the authors give guarantees in $\mathcal{W}_1$ - distance, we use $\mathcal{W}_2$. Our guarantee in $\mathcal{W}_1$-distances is $\mathcal{O}(\eta^{\frac{1}{4}})$, which is better than $\mathcal{O}(\eta^{\frac{1}{8}})$ in (Cheng et al., 2020).

In the case of SGD and SGLB (Ustimenko & Prokhorenkova, 2021), our work is the first to get estimates on discretization error in both cases of noise.

Table 3: Comparison of guarantees on discretization error in our work with the literature.

| | Noise | Reference | Rate |
|---|---|---|---|
| **SGLD** | Gaussian | (Muzellec et al., 2020) | $\mathcal{O}(\eta^{\frac{1}{4}})$ |
| | | Ours | $\mathcal{O}(\eta^{\frac{1}{4}})$ |
| | any | Ours | $\mathcal{O}(\eta^{\frac{1}{4}})$ |
| **SGD** | Gaussian | (Cheng et al., 2020) [1] | $\mathcal{O}(\eta^{\frac{1}{8}})$ |
| | | Ours | $\mathcal{O}(\eta^{\frac{3}{4}})$ |
| | any | (Cheng et al., 2020) [1] | $\mathcal{O}(\eta^{\frac{1}{8}})$ |
| | | Ours | $\mathcal{O}(\eta^{\frac{1}{2}})$ |
| **SGB** | any | Ours | $\mathcal{O}(\eta^{\frac{1}{2}})$ |
| **SGLB** | any | Ours | $\mathcal{O}(\eta^{\frac{1}{4}})$ |

[1] $\mathcal{W}_1$ - distance

## 2 PROBLEM SETUP AND ASSUMPTIONS

We consider the chain equation 1 and the diffusion equation 2 with $X_0 = Z_0 = x_0 \in \mathbb{R}^d$. Our ultimate goal is to produce an upper bound on $\mathcal{W}_2\big(\mathcal{L}(X_k), \mathcal{L}(Z_{k\eta})\big)$ for all $k \in \mathbb{N}$, where $\mathcal{L}$ here and after denotes the distribution of a random vector.

We assume that learning rate $\eta \in (0, 1]$ and $\gamma \in \{0, 1\}$ (see Table 1 for details). In the literature (especially on optimization), it is common to assume that the learning rate $\eta$ depends on the properties of $\nabla f$ (in our case $b$), e.g., typically that $\eta \sim L^{-1}$, where $L$ is a Lipschitz constant of $\nabla f$. Meanwhile, we can always renormalize the original problem to guarantee $\eta \in (0; 1]$. Next, let us list the assumptions concerning the terms of the chain equation 1.

**Assumption 1.** *There exists some constants $M_0 \geq 0$, $M_1 \geq 0$, $M_\epsilon \geq 0$, $\sigma_0 \geq 0$, $\sigma_1 \geq 0$, $\alpha > 0, \beta \in [0, 1]$ (see also Table 1 for special cases) such that*

• *the drift $b$ is $M_0$-Lipschitz, i.e. for $x, x' \in \mathbb{R}^d$*

$$\big\|b(x) - b(x')\big\| \leq M_0\big\|x - x'\big\|.$$

*For particular cases of optimization problems: minimization, and saddle point problems, this assumption means that the corresponding target functions have Lipschitz gradients, i.e., are smooth. Hence, we do not need other assumptions, such as the monotonicity of $b$ or, in particular, the convexity of $b = -\nabla f$.*

• *the biased drift $\delta$ is bounded, i.e. for all $k \geq 0$*

$$\big\|\delta_k\big\|^2 \leq M_1^2 \eta^{2\alpha}\big(1 + \big\|X_k\big\|^2\big).$$

- *the noise function $\epsilon$ is unbiased, has unit covariance, and "the central limit theorem" holds for it, i.e., for all $x \in \mathbb{R}^d$, $k \geq 0$ and $S \geq 1$*

$$\mathbb{E}_\epsilon \epsilon_k(x) = 0_d, \quad \mathbb{E}_\epsilon \epsilon_k \epsilon_k^{\mathrm{T}}(x) = I_d, \quad \mathcal{W}_2^2\Big(\mathcal{N}\big(0_d, SI_d\big), \mathcal{L}\Big(\sum_{k=0}^{S-1} \epsilon_k(x)\Big)\Big) \leq M_\epsilon^2 \eta^\beta.$$

*In the general case, noise $e$ can be arbitrary within the assumption of "the central limit theorem," which holds in particular for all noises that have uniformly bounded the fourth moment $\mathbb{E}\|\epsilon_k(x)\|^4$, see (Bonis, 2020). We also define $\chi_0 = 1$ if the noise $\epsilon_k$ is Gaussian, otherwise $\chi_0 = 0$, note that in the first case $M_\epsilon = 0$.*

- *the covariance coefficient $\sigma$ is symmetric, positive definite, uniformly elliptic, bounded and $M_0$-Lipschitz, i.e. for all $x, x' \in \mathbb{R}^d$*

$$\sigma_1 I_d \geq \sigma(x) = \sigma^{\mathrm{T}}(x) \geq \sigma_0 I_d, \quad \mathbb{E}_\epsilon \Big\| \sigma(x)\epsilon_k(x) - \sigma(x')\epsilon_k(x') \Big\|^2 \leq M_0^2 \|x - x'\|^2.$$

- *the covariance shift $\Delta$ is symmetric and bounded, i.e. for all $k \geq 0$*

$$\Delta_k = \Delta_k^{\mathrm{T}}, \quad \eta^\gamma \mathrm{Tr}\big(\Delta_k^2\big) \leq M_1^2 \eta^{2\alpha}\big(1 + \|X_k\|^2\big).$$

Let us also introduce the following notation for convenience: $M^2 = 2\max\big\{M_0^2, \|b(0_d)\|^2\big\} + 2\max\big\{M_0^2, d\sigma_1^2\big\} + 3\eta^{2\alpha}M_1^2$. One can note that such $M > 0$ gives

$$\|b(X_k)\|^2 + \mathrm{Tr}\Big(\sigma(X_k)\sigma^{\mathrm{T}}(X_k)\Big) + \|\delta_k\|^2 + \eta^\gamma \mathrm{Tr}\big(\Delta_k^2\big) \leq M^2\Big(1 + \|X_k\|^2\Big).$$

Moreover, we define the bound on the initial vector $x_0$: $R_0^2 = \max\{1; \|x_0\|^2\}$. Meanwhile, we do not assume the existence of some $X^*$ s.t. $X_k \to X^*$ because, in the general non-dissipative case, the chain equation 1 can diverge, i.e., $\|X_k\| \to \infty$.

## 3 MAIN RESULTS

### 3.1 CHAIN APPROXIMATION BY WINDOW COUPLING

The first thing to look into is the non-Gaussian noise. In this section, we construct a new auxiliary chain with Gaussian noise that approximates $S$-subsampled initial dynamic equation 1. The idea behind this approach is that we can use the central limit theorem for the noise function $\epsilon$ (Assumption 1), and note that by collecting a reasonably large $S$-batch, we can approximate our non-Gaussian noise by Gaussian.

We define $\epsilon_k^S(x) = \sum_{i=0}^{S-1} \epsilon_{Sk+i}(x)$. Since for $\epsilon$ "the central limit theorem" holds (Assumption 1), using this new "batched" $\epsilon$, we can introduce

$$\big(\sqrt{S}\zeta_k^S(x), \epsilon_k^S(x)\big) \sim \Pi_*\Big(\mathcal{N}(0_d, SI_d), \mathcal{L}\big(\epsilon_k^S(x)\big)\Big),$$

where $\Pi_*(\cdot, \cdot)$ is an optimal coupling for $\mathcal{W}_2$-distance. In fact, $\zeta$ has no closed form, but it is an auxiliary object, and we need only that it exists (by definition of $\mathcal{W}_2$) (Cavalletti & Huesmann, 2015). The only fact we strongly need follows from definitions of $\zeta_k^S$ and of optimal coupling – we know that $\zeta_S^k \sim \mathcal{N}(0_d, I_d)$. We need such $\zeta_S^k$ to introduce the coupled chain $(Y_{\overline{\eta}k}^X)$, which is closed to the chain $X_{Sk}$. The natural idea to construct is

$$Y_{\overline{\eta}(k+1)}^X = Y_{\overline{\eta}k}^X + \overline{\eta}\, b\big(Y_{\overline{\eta}k}^X\big) + \underbrace{\sqrt{\overline{\eta}\,\eta^\gamma}\, \sigma\big(Y_{\overline{\eta}k}^X\big)\zeta_k^S\big(X_{Sk}\big)}_{\text{coupling via noise}}, \tag{3}$$

where we additionally define subsampled learning rate $\overline{\eta} = S\eta$. The essence of this trick is quite simple – a non-Gaussian noise is subtracted from the original chain equation 1 and replaced by a Gaussian CLT approximation. This idea can show itself perfectly in the case of a $\mu$-strongly monotone operator $(-b)$ (which corresponds to strong convexity and strong dissipativity in the case $b = -\nabla f$): $\langle b(X) - b(Y), Y - X \rangle \geq \mu \|X - Y\|^2$. In particular, it is easy to bound $(Y_{\overline{\eta}k}^X)$ and $(X_{Sk})$ with monotonicity:

$$\mathbb{E}\big\|Y_{\overline{\eta}(k+1)}^X - X_{S(k+1)}\big\|^2 = \mathbb{E}\big\|Y_{\overline{\eta}k}^X + \overline{\eta}b\big(Y_{\overline{\eta}k}^X\big) + \ldots - X_{Sk} - \eta\sum_{i=0}^{S-1} b(X_{Sk+i}) + \ldots\big\|^2$$

$$
\begin{aligned}
&\approx\quad \mathbb{E}\big\|Y_{\overline{\eta}k}^X + \overline{\eta}b\big(Y_{\overline{\eta}k}^X\big) - X_{Sk} - \overline{\eta}b(X_{Sk}) + \dots \big\|^2 \\
&=\quad \mathbb{E}\big\|Y_{\overline{\eta}k}^X - X_{Sk}\big\|^2 + 2\overline{\eta}\mathbb{E}\langle b\big(Y_{\overline{\eta}k}^X\big) - b(X_{Sk}), Y_{\overline{\eta}k}^X - X_{Sk}\rangle + \dots \\
&\leq\quad (1 - \mu\overline{\eta})\mathbb{E}\big\|Y_{\overline{\eta}k}^X - X_{Sk}\big\|^2 + \dots.
\end{aligned}
$$

If we choose $\overline{\eta} < \mu^{-1}$, then we have a geometric convergence. It allows us to obtain the bound on the difference that does not diverge in time. *But this idea does not work in our setting because nothing like (strong) monotonicity/convexity/dissipativity is required to hold*. Without such a term, those two chains mainly diverge from each other in general, as no structure pulls them closer to each other. This issue can be alleviated. We need to add one more term. In particular, we inject $\overline{\eta}L\big(X_{Sk} - Y_{\overline{\eta}k}^X\big)$ with some constant $L > 0$ (which we define later) into equation 3. Thus, instead of equation 3 we define the coupled chain $Y_{\overline{\eta}k}^X$ using arbitrary fixed constant $L \geq 0$:

$$
Y_{\overline{\eta}(k+1)}^X = Y_{\overline{\eta}k}^X + \overline{\eta}b(Y_{\overline{\eta}k}^X) + \sqrt{\overline{\eta}\eta^\gamma}\sigma(Y_{\overline{\eta}k}^X)\zeta_k^S\big(X_{Sk}\big) \underbrace{-L\overline{\eta}(Y_{\overline{\eta}k}^X - X_{Sk})}_{\text{Window coupling}}. \tag{4}
$$

This additional coupling effectively emulates "monotonicity"/"convexity"/"dissipativity":

$$
\begin{aligned}
\mathbb{E}\big\|Y_{\overline{\eta}(k+1)}^X - X_{S(k+1)}\big\|^2 &=\quad \mathbb{E}\big\|Y_{\overline{\eta}k}^X + \overline{\eta}L\big(X_{Sk} - Y_{\overline{\eta}k}^X\big) + \dots - X_{Sk} + \dots\big\|^2 \\
&=\quad \mathbb{E}\big\|Y_{\overline{\eta}k}^X - X_{Sk}\big\|^2 + 2\overline{\eta}L\mathbb{E}\langle X_{Sk} - Y_{\overline{\eta}k}^X, Y_{\overline{\eta}k}^X - X_{Sk}\rangle + \dots \\
&\leq\quad (1 - 2\overline{\eta}L)\mathbb{E}\big\|Y_{\overline{\eta}k}^X - X_{Sk}\big\|^2 + \dots.
\end{aligned} \tag{5}
$$

Physical interpretation of the additional term $L\overline{\eta}(Y_{\overline{\eta}k}^X - X_{Sk})$ is straightforward: in order to force $Y_{\overline{\eta}k}^X$ and $X_{Sk}$ to be close to each other we "forget" a portion of $Y_{\overline{\eta}k}^X$ and replace it with a portion of $X_{Sk}$. Normal noise appearing in $Y_{\overline{\eta}k}^X$ makes the distribution of $Y_{\overline{\eta}k}^X$ "regular" and $-\overline{\eta}LY_{\overline{\eta}k}^X$ term acts like $L_2$-regularization for $Y_{\overline{\eta}k}^X$ which additionally imposes regularity on $Y_{\overline{\eta}k}^X$, which was absent in $X$ due to the non-Gaussian noise and absence of the regularization.

On the other hand, the purpose of introducing $Y_{\overline{\eta}k}^X$ is to create a Gaussian chain that approximates the original non-Gaussian $X_k$. But $L\overline{\eta}(Y_{\overline{\eta}k}^X - X_{Sk})$ at first glance destroys the normality of $Y_{\overline{\eta}k}^X$. We suggest to look at Window coupling $L\overline{\eta}(Y_{\overline{\eta}k}^X - X_{Sk})$ as a biased part of drift. Then the noise in the $Y_{\overline{\eta}k}^X$ remains Gaussian. Nevertheless, this will give us additional problems in the future that will have to be solved.

Even though reasoning equation 5 gives the basic idea of convergence of $\mathbb{E}\big\|X_{Sk} - Y_{\overline{\eta}k}^X\big\|^2$, we hide a lot under the dot sign, and these missing terms can have a bad effect. In particular, if we try to bound $\mathbb{E}\big\|X_{Sk} - Y_{\overline{\eta}k}^X\big\|^2$, then we would immediately obtain divergence since the expected squared norm of $X_k$ can be unbounded (as we noted in Section 2). Such a norm appears multiplicative in bias-related and covariance-related terms (e.g., $\delta$ and $\Delta$ in Assumption 1). This suggests that if we temper the difference $\mathbb{E}\big\|X_{Sk} - Y_{\overline{\eta}k}^X\big\|^2$ on the bound of the worst-case norm $R^2(t) \geq \max\big\{1; \max_{\eta k \leq t} \mathbb{E}\|X_k\|^2\big\}$, then we effectively cancel out terms with $\mathbb{E}\|X_{Sk}\|^2$ and obtain "uniform" in $k\overline{\eta} \leq t$ bound for any fixed horizon $t > 0$.

**Lemma 1.** *Let Assumption 1 holds. If* $L \geq 1 + M_0 + M + 2M_0^2 + M^2 + M^2M_\epsilon^2$ *and* $S\eta \leq 1$*, then for any* $t > 0$*:*

$$
\Delta_t^S = \max_{\overline{\eta}k' \leq t} \frac{\mathbb{E}\big\|X_{Sk'} - Y_{\overline{\eta}k'}^X\big\|^2}{R^2(t)} = \mathcal{O}\left(\eta^{2\alpha} + \frac{\eta^{\gamma+\beta}}{S}(1 - \chi_0) + \overline{\eta}\right),
$$

*where* $R^2(t) \geq \max\big\{1; \max_{\eta k \leq t} \mathbb{E}\|X_k\|^2\big\}$*.*

Here and after, in the main part, we use $\mathcal{O}$-notation to hide constants that do not depend on $\eta$, $k$, and $t$ for simplicity. The full statements are given in Appendix. Since the chain $Y_{\overline{\eta}k}^X$ is artificially introduced by us, we can vary $S$, in particular, optimize the estimate from Lemma 1

**Corollary 1.** *Under the conditions of Lemma 1. If we take* $S = \eta^{-\frac{1-\beta}{2}}(1 - \chi_0) + \chi_0 \geq 1$*, then it holds that:*

$$
\Delta_t^S \leq \mathcal{O}\big(\eta^{2\theta}\big) \quad with \quad \theta = \min\left\{\alpha; \frac{(\gamma+1)(1+\chi_0) + (\gamma+\beta)(1-\chi_0)}{4}\right\}.
$$

## 3.2 NAIVE INTERPOLATION OF THE APPROXIMATION

As mentioned earlier, if we put $\overline{\eta}L(X_{Sk} - Y^X_{\overline{\eta}k})$ to a biased drift, we can look at the chain equation 4 as a chain with Gaussian noise, i.e., we can rewrite equation 4 in the same form as equation 1:

$$Y^X_{\overline{\eta}(k+1)} = Y^X_{\overline{\eta}k} + \overline{\eta}(b(Y^X_{\overline{\eta}k}) + g^S_{\overline{\eta}k}) + \sqrt{\overline{\eta}}\sqrt{\eta^\gamma}\sigma(Y^X_{\overline{\eta}k})\zeta^S_k,$$

where we introduce $g^S_{\overline{\eta}k} = L(X_{Sk} - Y^X_{\overline{\eta}k})$.

For a chain with Gaussian noise, there are well-known tricks for relating them to some diffusion process driven by the Brownian motion process $W_t$. In most of the works, e.g., (Raginsky et al., 2017), on Langevin processes from Table 2, one embeds a process with Gaussian noise into diffusion without any hesitation due to normality of the noise, which allows one to think that the noises are increments of the Brownian motion process $W_t$ and intermediate points are obtained through naive mid-point interpolation (e.g., by replacing $\overline{\eta}$ with $\delta t$ for $\delta t \in [0, \overline{\eta}]$ allows to define $Y^X_{\overline{\eta}k+\delta t}$). Since $\zeta^S_k \sim \mathcal{N}(0_d, I_d)$, we consider $\zeta^S_k = \sqrt{\overline{\eta}^{-1}}(W_{(k+1)\overline{\eta}} - W_{k\overline{\eta}})$. One can also complete the notation for $g^S_t = g^S_{[t/\overline{\eta}]\overline{\eta}}$ for all $t$ (before we defined it only for $t = \overline{\eta}k$ with integer $k$). Then we can define the following diffusion:

$$\mathrm{d}Y_t = \big(b(Y_t) + (g^*_t + g^S_t)\big)\mathrm{d}t + \sqrt{\eta^\gamma}\big(\sigma(Y_t) + \Sigma^*_t\big)\mathrm{d}W_t,$$

with new notation $g^*_t = b(Y^X_{[t/\overline{\eta}]\overline{\eta}}) - b(Y_t)$ and $\Sigma^*_t = \sigma(Y^X_{[t/\overline{\eta}]\overline{\eta}}) - \sigma(Y_t)$.

The diffusion $Y_t$ has a "ladder" drift (since $b(Y_t) + (g^*_t + g^S_t)$ changes only in $t = k\eta$), thereby $Y_t$ mimicks the chain $Y^X_{\overline{\eta}k}$. If there were no noise terms in the diffusion and in the chain, $\mathrm{d}Y_t$ would fully coincide with $Y^X_{\overline{\eta}k}$. Nevertheless, we have noise components, and in diffusion, it is provided by the Brownian process $W_t$ and is continuous in time, unlike the Gaussian, but "discrete" noise term in $Y^X_{\overline{\eta}k}$ (depending on $\zeta^S_k$). We need to estimate the differences in these noises. This is the idea of estimating the difference between $Y^X_{\overline{\eta}k}$ and $\mathrm{d}Y_t$. We follow a similar path of proving as (Raginsky et al., 2017), but we also consider that we now face non-dissipativity. In particular, the next lemma gives a bound that includes $R^2(k\eta)$, which is not the case of (Raginsky et al., 2017).

**Lemma 2.** *Let Assumption 1 holds. Then for any $k \in \mathbb{N}_0$:*

$$\sup_{t \le k\eta} \mathbb{E}\big\|Y^X_{[t/(\overline{\eta})]\overline{\eta}} - Y_t\big\|^2 = \mathcal{O}\left(\overline{\eta}^{1+\gamma}R^2(k\eta)\right),$$

*where $R^2(k\eta) \ge \max\left\{1; \max_{k' \le k} \mathbb{E}\|X_{k'}\|^2\right\}$.*

From the definition of $\mathcal{W}_2$, we immediately have.

**Corollary 2.** *Under the conditions of Lemma 2, for any time horizon $t = k\eta \ge 0$ it holds:*

$$\mathcal{W}^2_2(\mathcal{L}(Y^X_{[t/(\overline{\eta})]\overline{\eta}}), \mathcal{L}(Y_t)) \le \sup_{t \le k\eta} \mathbb{E}\big\|Y^X_{[t/(\overline{\eta})]\overline{\eta}} - Y_t\big\|^2 = \mathcal{O}\left(\overline{\eta}^{1+\gamma}R^2(k\eta)\right).$$

At this point, we linked the initial chain $X_k$ to the auxiliary chain $Y_{\overline{\eta}k}$, and then the chain $Y_{\overline{\eta}k}$ to the diffusion $Y_t$. However, the diffusion $Y_t$ differs from the target diffusion $Z_t$ we aim for. Therefore, we are left to link diffusions $Y_t$ and $Z_t$ - we will spend the following three subsections on this.

## 3.3 COVARIANCE CORRECTED INTERPOLATION

The problem with the relation of $Y_t$ and $Z_t$ lies primarily in the fact that these two diffusions have different covariance coefficients ($\big(\sigma(Y_t) + \Sigma^*_t\big)$ and $\sigma(Z_t)$). In that case, the Girsanov theorem states that diffusions with *different* covariances are singular to each other. Meanwhile, if the covariance coefficients are the same, then the Girsanov theorem (Liptser & Shiryaev, 2001) gives a more optimistic answer to the possibilities of connecting such diffusions. Then our goal now is to deal with $\Sigma^*_t$ in $Y_t$.

It is proposed to build another auxiliary diffusion. We want the problems of the covariance coefficient in $Y_t$ to move elsewhere, e.g., to drift, in the new chain $Z^Y_t$. We have already done a similar trick with the coupling of $X_k$ and $Y^X_{\overline{\eta}k}$ in Section 3.1. Then let us consider the following diffusion:

$$\mathrm{d}Z^Y_t = (b(Z^Y_t) + g^S_t)\mathrm{d}t + \underbrace{L_1(Y_t - Z^Y_t)\mathrm{d}t}_{\text{Window coupling}} + \sqrt{\eta^\gamma}\sigma(Z^Y_t)\mathrm{d}W_t,$$

where $L_1$ we will define later. This coupling helps to eliminate not only $\Sigma_t^*$ but also $g_t^*$. The basic idea is that $\Sigma_t^*$ and $g_t^*$ are differences of $\sigma$ and $b$, respectively. At the same time, $\sigma$ and $b$ are Lipschitz, which means that we can bound $\Sigma_t^*$ and $g_t^*$ via the norm of the argument difference. Therefore, if we choose $L_1$ large enough, we compensate bounds on $\Sigma_t^*$ and $g_t^*$. In particular, the following statement holds.

**Lemma 3.** *Let Assumption 1 holds. Then for $L_1 \geq 2M_0 + 4M_0\eta^\gamma$ and any time horizon $t = k\eta \geq 0$:*

$$\mathcal{W}_2^2(\mathcal{L}(Y_t), \mathcal{L}(Z_t^Y)) \leq \sup_{t' \leq t} \mathbb{E}\left\|Y_{[t'/(\overline{\eta})]\overline{\eta}}^X - Y_{t'}\right\|^2.$$

From Lemma 2, we immediately have.

**Corollary 3.** *Let Assumption 1 holds. Then for any time horizon $t = k\eta \geq 0$:*

$$\mathcal{W}_2^2(\mathcal{L}(Y_t), \mathcal{L}(Z_t^Y)) = \mathcal{O}\left(\overline{\eta}^{1+\gamma} R^2(k\eta)\right),$$

*where $R^2(k\eta) \geq \max\left\{1; \max_{k' \leq k} \mathbb{E}\|X_{k'}\|^2\right\}$.*

To complete the proof, it remains to relate two diffusions $Z_t$ and $Z_t^Y$ in terms of $\mathcal{W}_2$.

### 3.4 ENTROPY BOUND FOR DIFFUSION APPROXIMATION

With the introduction of the new notation $G_t^S = g_t^S - L_1(Z_t^Y - Y_t)$, one can rewrite $Z_t^Y$ as follows:

$$dZ_t^Y = (b(Z_t^Y) + G_t^S)dt + \sqrt{\eta^\gamma}\sigma(Z_t^Y)dW_t.$$

At the moment, the only difference between $Z_t$ and $Z_t^Y$ is the presence of $G_t^S$ in the drift of $Z_t^Y$. Finally, when the correlation coefficients are equal, we are close to using the Girsanov theorem (Liptser & Shiryaev, 2001). However, its classical version requires that both diffusions are Markovian. Unfortunately, this is not our case because of $G_t^s$. In particular, $g_t^s$ (part of $G_t^s$) was defined in Section 5 as follows: $g_{\overline{\eta}k}^S = L(X_{Sk} - Y_{\overline{\eta}k}^X)$ for $k \in \mathbb{N}_0$ and $g_{[t/\overline{\eta}]\overline{\eta}}^S$ for all others $t \geq 0$. It turns out that for $t \neq k\overline{\eta}$, $g_t^S$ depends on the nearest reference point $[t/\overline{\eta}]\overline{\eta}$, which immediately violates the Markovian property. The same problem arises in (Raginsky et al., 2017). The way out of this issue is to prove a new version of the Girsanov theorem. The paper (Raginsky et al., 2017) considers the dissipative case where additionally $\sigma(\cdot)$ is constant and independent of the current state. We have even more general and complex versions of the Girsanov theorem without dissipativity and for state-dependent $\sigma(\cdot)$.

**Theorem 1** (One-time Girsanov formula for mixed Ito/adapted coefficients). *Assume that $(G_t)_{t \geq 0}$ is a $(W_t)_{t \geq 0}$-adapted process with an integratable by $t \geq 0$ second moment. Consider two SDEs ran using two independent Brownian processes:*

$$dZ_t = b(Z_t)dt + G_t dt + \sigma(Z_t)dW_t,$$
$$dZ_t^* = b(Z_t^*)dt \qquad\quad + \sigma(Z_t^*)d\widetilde{W}_t.$$

*Let $\sigma_0 > 0$ be the minimal possible eigenvalue of $\sigma(x)$. Then we have that for any time horizon $T \geq 0$ the following bound holds:*

$$D_{\mathrm{KL}}\left(\mathcal{L}(Z_T)\big\|\mathcal{L}(Z_T^*)\right) \leq \sigma_0^{-2} \int_0^{\mathrm{T}} \mathbb{E}\|G_t\|^2 dt < \infty.$$

The result of Theorem 1 is worse than the classical version, which is predictably the price of generalization. This theorem can be considered as a stand-alone contribution of the paper. Let us apply it to our case with $Z_t$ and $Z_t^Y$.

**Corollary 4.** *Let Assumption 1 holds. Then for any time horizon $t = \eta k \geq 0$:*

$$D_{\mathrm{KL}}\left(\mathcal{L}(Z_{k\eta}^Y)\big\|\mathcal{L}(Z_{k\eta})\right) = \mathcal{O}\left(\eta^{-\gamma} k\eta e^{\mathcal{O}(k\eta)} \eta^{2\theta}\right),$$

*where $\theta = \min\left\{\alpha; \frac{(\gamma+1)(1+\chi_0) + (\gamma+\beta)(1-\chi_0)}{4}\right\}$.*

### 3.5 EXPONENTIAL INTEGRABILITY

At the moment, our modification of the Girsanov theorem gives the estimates on the relation between $Z_{k\eta}$ and $Z_{k\eta}^Y$ only in terms of KL-divergence. Meanwhile, our ultimate goal is to bound the $\mathcal{W}_2$-distance. Therefore, we need to connect the estimates for KL-divergence and $\mathcal{W}_2$-distance. Let us use the auxiliary result from (Bolley & Villani, 2005) to solve this issue. In more details, for two distributions $p_1$ and $p_2$, we have

$$\mathcal{W}_2(p_1, p_2) \leq \mathcal{C}_{\mathcal{W}}(p_2) \left( \left( D_{\mathrm{KL}}(p_1 || p_2) \right)^{\frac{1}{2}} + \left( \frac{D_{\mathrm{KL}}(p_1 || p_2)}{2} \right)^{\frac{1}{4}} \right), \tag{6}$$

where $\mathcal{C}_{\mathcal{W}}(p_2) > 0$ relates entropy $D_{\mathrm{KL}}(\cdot || p_2)$ with $\mathcal{W}_2(\cdot, p_2)$. The main challenge here is to find the bound on the constant $\mathcal{C}_{\mathcal{W}}(p_2)$ with $p_2 = \mathcal{L}(Z_{k\eta})$ for all $k \in \mathbb{N}$ or a slightly more general result with $p_2 = \mathcal{L}(Z_t)$ for all $t \geq 0$. Related results are available in the literature (Raginsky et al., 2017), but were obtained in special cases: under dissipative conditions and with specific $\gamma$ equal to 0. The challenge of finding $\mathcal{C}_{\mathcal{W}}^2$ in the non-dissipative case for arbitrary $\gamma$ is solved in the next theorem.

**Lemma 4.** *Let Assumption 1 holds. Then for any time horizon $t \geq 0$:*

$$\mathcal{C}_{\mathcal{W}}(\mathcal{L}(Z_t)) = \mathcal{O}\left( \eta^{\frac{\gamma}{2}} e^{\mathcal{O}(t)} \right).$$

The presence of factor $\eta^{\frac{\gamma}{2}}$ for $\gamma \neq 0$ is not only novel but also crucial. In the previous section, we already encountered this $\eta^{-\gamma}$. In the final estimate, these two factors will cancel each other out.

### 3.6 FINAL RESULT

It remains to combine all the results obtained above. In particular, we require Corollaries 1, 2, 3 and 4, as well as Lemma 4. It is important to note that in Corollary 1, $S = \eta^{-\frac{1-\beta}{2}}(1 - \chi_0) + \chi_0 \geq 1$ has already been chosen, it needs to be substituted to other results.

**Theorem 2.** *Let Assumption 1 hold. Then for all $k \in \mathbb{N}_0$:*

$$\mathcal{W}_2(\mathcal{L}(X_{k'}), \mathcal{L}(Z_{k'\eta})) = \mathcal{O}\left( \left(1 + (k'\eta)^{\frac{1}{2}}\right) e^{\mathcal{O}(k'\eta)} \eta^\theta + (k'\eta)^{\frac{1}{4}} e^{\mathcal{O}(k'\eta)} \eta^{\frac{\theta}{2} + \frac{\gamma}{4}} \right),$$

*where $\theta = \min\left\{ \alpha; \frac{(\gamma+1)(1+\chi_0)+(\gamma+\beta)(1-\chi_0)}{4} \right\}$.*

All (exponentially) growing factors in Theorem 2 depend only on horizon $T \geq k\eta$, which is assumed to be fixed a priory. Thus, if we consider convergence on the interval $t \in [0, T]$, then those factors are essentially constants, independent from $\eta^{-1}$. Though, compared to the work on SGLD (Raginsky et al., 2017) that relies on dissipativity assumption, those constants grow exponentially, which limits the applicability of the results for such problems as sampling from the invariant measure. Since our results do not rely on dissipativity/convexity assumptions, having exponential dependence on the horizon is unavoidable in the general case (Alfonsi et al., 2014). Putting the horizon $T$ fixed, we obtain that $\mathcal{W}_2(\mathcal{L}(X_k), \mathcal{L}(Z_{k\eta})) = \mathcal{O}\left( \eta^\theta + \eta^{\frac{\theta}{2} + \frac{\gamma}{4}} \right)$. From this, one can find estimates for the different cases from Table 1. As an example:

**Corollary 5** (SGD with Gaussian noise). *Under the conditions of Theorem 2, if $\chi_0 = 1$, $\gamma = 1$, $\alpha \geq 1$, then $\theta = 1$ and $\mathcal{W}_2(\mathcal{L}(X_k), \mathcal{L}(Z_{k\eta})) = \mathcal{O}\left( \eta^{\frac{3}{4}} \right)$.*

## 4 CONCLUSION

This work considers a broad class of Markov chains, Ito chains. Moreover, we introduce the most general assumptions on the terms of our Ito chain. In particular, we assume that the noise can be almost arbitrary but not independent and normal. In this setting, we estimate $\mathcal{W}_2$ between the chain and the corresponding diffusion. Our estimates cover a large variety of special cases: in some cases replicating the state-of-the-art results, in others improving them, and in some cases being pioneering.

ACKNOWLEDGMENTS

This research of A. Beznosikov has been supported by The Analytical Center for the Government of the Russian Federation (Agreement No. 70-2021-00143 dd. 01.11.2021, IGK 000000D730321P5Q0002).

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

## A    PRELIMINARIES

Primary we are going to work with Wasserstein-2 distance that would be defined a few lines later. Wasserstein-2 distance metricize topology of weak convergence plus the convergence of the second moments which are desirable properties for an algorithm to have and allow us to consider functions that grow at most quadratically compared to uniformly bounded if we considered just weak convergence without second moments as done in the original SGLB paper.

Now let's define the metric. Denote by

$$\Pi(\nu, \mu) := \big\{\pi(\mathrm{d}x_1, \mathrm{d}x_2) \big| \int_{\mathbb{R}^d} \pi(\mathrm{d}x_1, \cdot) = \mu \wedge \int_{\mathbb{R}^d} \pi(\cdot, \mathrm{d}x_2) = \nu \big\}$$

– the set of all possible couplings of measures $\mu(\mathrm{d}x)$ and $\nu(\mathrm{d}x)$. Then define:

$$\mathcal{W}_2(\nu, \mu) = \Big( \inf_{\pi \in \Pi(\nu, \mu)} \mathbb{E}_\pi \big| x_1 - x_2 \big|^2 \Big)^{\frac{1}{2}}$$

– Wasserstein-2 distance between measures $\nu(\mathrm{d}x)$ and $\mu(\mathrm{d}x)$. The unique coupling achieving the infinum we would denote as $\pi_* = \Pi_*(\nu, \mu)$.

The following tautological statement would be useful when dealing with the Wasserstein-2 metric:

**Statement 1.** *Assume that we are given two random variables $x_1 \sim \nu, x_2 \sim \mu$ defined on common probability space with measure $\pi$, then Wasserstein-2 distance between $\nu$ and $\mu$ is bounded by $L_2$-distance between $x_1$ and $x_2$:*

$$\mathcal{W}_2(\nu, \mu) \leq \sqrt{\mathbb{E}_\pi \big| x_1 - x_2 \big|^2}$$

Moreover, we are going to use Kullback-Liebner divergence $D_{\mathrm{KL}}(\nu\|\mu)$ between measures $\nu$ and $\mu$ such that $\nu \ll \mu$ (absolute continious). In that case by Random-Nikodym theorem there exists $\mu$-measurable mapping $\frac{\mathrm{d}\nu}{\mathrm{d}\mu}$ that is non-negative $\mu$-almost everywhere such that $\mathbb{E}_\mu \frac{\mathrm{d}\nu}{\mathrm{d}\mu} = 1$ and $\mathbb{E}_\nu f = \mathbb{E}_\mu \big(f\frac{\mathrm{d}\nu}{\mathrm{d}\mu}\big)$ for any bounded measurable $f$. The mapping $\frac{\mathrm{d}\nu}{\mathrm{d}\mu}$ is called Radom-Nikodym derivative of $\nu$ with respect to $\mu$. Having $\frac{\mathrm{d}\nu}{\mathrm{d}\mu}$ we define Kullback-Liebner divergence as:

$$D_{\mathrm{KL}}(\nu\|\mu) \triangleq \mathbb{E}_\nu \log \frac{\mathrm{d}\nu}{\mathrm{d}\mu} = \int_{\mathrm{supp}\,\mu} \frac{\mathrm{d}\nu}{\mathrm{d}\mu} \log \frac{\mathrm{d}\nu}{\mathrm{d}\mu} \mathrm{d}\mu \geq 0$$

The following statement is known as data-processing inequality:

**Statement 2.** *(DPI; Data processing inequality, see Beaudry & Renner (2012)) Let $T$ we some $\mu$-measurable mapping and $\nu \ll \mu$ – measures. Let $T_{\#}\mu \triangleq \mu(T^{-1}(\cdot))$ – push-forward measure under mapping $T$. Then we have an inequality:*

$$D_{\mathrm{KL}}\big(T_{\#}\nu\|T_{\#}\mu\big) \leq D_{\mathrm{KL}}\big(\nu\|\mu\big)$$

We are going to apply DPI to $\mu = \mathcal{L}(X_t : t \leq k\eta)$ and $\nu = \mathcal{L}(Z_t : t \leq k\eta)$ for a certain diffusion processes $(X_t)_{t \geq 0}$ and $(Z_t)_{t \geq 0}$ with the same diffusion coefficient (to ensure absolute continuity since otherwise the measures are mutually singular due to Girsanov theorem) so that Radom-Nykodym derivative exists and henceforth $D_{\mathrm{KL}}(\nu\|\mu) < \infty$. As push-forward map $T$ we can consider $T\big((Z_t)_{0 \leq t \leq k\eta}\big) \triangleq Z_{k\eta}$. Then $T$ is $\mu$-measurable with the property $T_{\#}\mu = \mathcal{L}(X_{k\eta})$ and therefore we obtain:

$$D_{\mathrm{KL}}\big(\mathcal{L}(Z_{k\eta})\|\mathcal{L}(X_{k\eta})\big) \leq D_{\mathrm{KL}}\big(\nu\|\mu\big)$$

DPI can be trivially proved: consider minimal $\sigma$-algebra generated by pre-image of the given $\sigma$-algebra under mapping $T$. We denote such $\sigma$-algebra as $\mathcal{F}_T$. After noting that $T_{\#}\nu \ll T_{\#}\mu$ if $\nu \ll \mu$ we consider $\mathcal{F}_T$-measurable random-variable $\mathbb{E}\big(\frac{\mathrm{d}\nu}{\mathrm{d}\mu}\big|\mathcal{F}_T\big)$. Then $\frac{\mathrm{d}T_{\#}\nu}{\mathrm{d}T_{\#}\mu}$ exists and $\frac{\mathrm{d}T_{\#}\nu}{\mathrm{d}T_{\#}\mu} \equiv \mathbb{E}\big(\frac{\mathrm{d}\nu}{\mathrm{d}\mu}\big|\mathcal{F}_T\big)$ $\mu$-almost everywhere. Then conditional Jensen inequality applied to KL implies DPI. In the work we are going to deal with a wide range of sequences in the form:

$$r_{k+1} \leq (1 - \eta\sigma_1)r_k + \eta\sigma_2 \tag{7}$$

with $r_0 = 0$, $0 < \eta\sigma_1 < 1$ and $\sigma_2 \geq 0$. Iterating such sequence immediately yields the formula:

$$r_k \leq (1 - (1 - \eta\sigma_1)^{k+1})\frac{\sigma_2}{\sigma_1} \leq \frac{\sigma_2}{\sigma_1} \tag{8}$$

Clearly, in order for the bound to be sound one needs $\sigma_2 \ll \sigma_1$.

The following theorem plays significant role in the proofs of our main results:

**Theorem 3.** *(Girsanov, see Theorems 12.1, 12.2 on p. 368-370 in Baldi & Baldi (2017)) Assume that we are given two SDEs in Ito form:*

$$\mathrm{d}X_{i,t} = b_i(X_{i,t}, t)\mathrm{d}t + \sigma_i(X_{i,t}, t)\mathrm{d}W_t$$

*for $i \in \{1, 2\}$ and where $W_t$ is standart Wiener process valued in $\mathbb{R}^d$. Denote by $\mu_i \triangleq \mathcal{L}(X_{i,t} : 0 \leq t \leq T)$ for some fixed $T > 0$. Assume that functions $b_i(x, t), \sigma_i(x, t)$ are Lipshitz continious in $x$ and $\sigma_i(x, t) = \sigma_i^T(x, t) \geq \sigma_T I_d$ almost everywhere in $x$ for some constant $\sigma_T > 0$ and for every $t \in [0, T]$. Then $\mu_2 \ll \mu_1$ if and only if $\sigma_1(x, t) \equiv \sigma_2(x, t) \equiv \sigma(x, t)$ almost everywhere in $x \in \mathbb{R}^d$ $\forall t(0 \leq t \leq T)$ and the Radom-Nykodym derivative is given by:*

$$\frac{\mathrm{d}\mu_2}{\mathrm{d}\mu_1} = e^{Z_T((X_1)_{t \leq T})}$$

*Where we denote $\Delta b(x, t) \triangleq \{\sigma(x, t)\}^{-1}(b_2(x, t) - b_1(x, t))$ and:*

$$Z_t(X) \triangleq \underbrace{\int_0^t \langle \Delta b(X_s, s), \mathrm{d}W_s \rangle_2}_{\text{Ito Matringale}} - \int_0^t \left|\Delta b(X_s, s)\right|^2 \mathrm{d}s\,,$$

*where $X = (X_s)_{s \leq t}$ any vector valued continuous path defined over $[0, t]$.*

Moreover, the following result due to Gyöngy (1986) would be needed within Girsanov theorem:

**Theorem 4.** *(Gyongy Gyöngy (1986)) Assume that we are given the following SDE:*

$$\mathrm{d}X_t = B_t\mathrm{d}t + \sigma_t\mathrm{d}W_t$$

*where $(B_t)_{t \geq 0}, (\sigma_t)_{t \geq 0}$ are some predictable w.r.t. $(W_t)_{t \geq 0}$ processes. Let $(X_t)_{t \geq 0}$ be a solution. Then $\mathcal{L}(X_t) = \mathcal{L}(\widetilde{X}_t) \forall t \geq 0$ where $(\widetilde{X}_t)_{t \geq 0}$ is a solution to Ito SDE:*

$$\mathrm{d}\widetilde{X}_t = g(\widetilde{X}_t, t)\mathrm{d}t + \sigma(\widetilde{X}_t, t)\mathrm{d}W_t$$

*where $g(x, t) \triangleq \mathbb{E}(B_t|X_t = x)$ and $\sigma(x, t) \triangleq \sqrt{\mathbb{E}(\sigma_t\sigma_t^{\mathrm{T}}|X_t = x)}$.*

We also consider the following construction: for a chain $(X_k)_{k \in \mathbb{Z}_+}$ in the form:

$$X_{k+1} = X_k + \eta B_k + \mathcal{N}(0_d, \eta\sigma^2 I_d)$$

where $(B_k)_{k \in \mathbb{Z}_+}$ is $(X_k)_{k \in \mathbb{Z}_+}$-predictable, we define the process $\overline{X}_t$:

$$\mathrm{d}\overline{X}_t = -\overline{B}_t\mathrm{d}t + \sigma\mathrm{d}W_t$$

where $\overline{B}_t \triangleq \sum_{k=0}^{\infty} B_k \mathbf{1}_{t \in [\eta k, \eta(k+1))}$. Then we obtain that:

$$\mathcal{L}(X_k : k \in \mathbb{Z}_+) = \mathcal{L}(\overline{X}(k\eta) : k \in \mathbb{Z}_+).$$

Combining with the result due to Gyongy we, moreover, obtain:

$$\mathcal{L}(\overline{X}_t) = \mathcal{L}(\widetilde{X}_t) \forall t \geq 0,$$

where $(\widetilde{X}_t)_{t \geq 0}$ satisfies Ito SDE:

$$\mathrm{d}\widetilde{X}_t = \mathbb{E}(\overline{B}_t|\overline{X}_t = \widetilde{X}_t)\mathrm{d}t + \sigma\mathrm{d}W_t$$

We call such processes $(\overline{X}_t)_{t \geq 0}$ and $(\widetilde{X}_t)_{t \geq 0}$ as interpolation of the chain $(X_k)_{k \in \mathbb{Z}_+}$ into the continious process. The core idea of interpolation is to obtain a bound on $D_{\mathrm{KL}}$ between the chain' law and some other chain' law by interpolating them into continuous processes in Ito form (recall that $(\widetilde{X}_t)_{t \geq 0}$ is in Ito SDE form) and then by using Girsanov theorem followed by DPI to switchback to the chains.

**Theorem 5.** *(Deterministic Bihari-LaSalle inequality Bihari (1956)). Let $H$ be constant, $x(t) \geq 0$ a càdlàg function for $t \geq 0$ and $A(t) \geq 0$ a non-decreasing function càdlàg function with $A(0) = 0$. Let $w(x) > 0$ for $x > 0$ be continuous non-decreasing function on $\mathbb{R}_{\geq 0}$. Let $W(v) = \int_C^v \frac{1}{w(x)} \mathbf{d}x$ for some $C > 0$. If function $x(t)$ satisfies:*

$$x(t) \leq \int_{0+}^t w(x(t))\mathbf{d}A(t) + H \forall t \in [0, T]$$

*for some $T > 0$ and $H > 0$ and if $W(H) + A(t)$ is in domain of $W^{-1}$ then:*

$$x(t) \leq W^{-1}(W(H) + A(t)) \forall t \in [0, T]$$

## B MISSING PROOFS

### B.1 USEFUL LEMMAS

**Lemma 5.** *Let Assumption 1 holds. Then for any $k \in \mathbb{N}_0$:*

$$1 + \mathbb{E}\|X_k\|^2 \leq e^{Ck\eta}\big(1 + \|x_0\|^2\big) \triangleq R^2(k\eta),$$

*where $C = 8(1 + M^2)$.*

*Proof.* Using the definition of equation 1, we get

$$\|X_{k+1}\|^2 = \|X_k\|^2 + \eta^2\big\|b(X_k) + \delta_k\big\|^2 + \eta^{1+\gamma}\Big\|\big(\sigma(X_k) + \Delta_k\big)\epsilon_k(X_k)\Big\|^2 + 2\eta\langle X_k, b(X_k) + \delta_k\rangle + \xi_k,$$

where $\xi_k = 2\sqrt{\eta^{1+\gamma}}\langle(\sigma(X_k) + \Delta_k)\epsilon_k(X_k), X_k + \eta(b(X_k) + \delta_k)\rangle$. With unbiasedness of $e_k(X_k)$, we have that $\mathbb{E}[\xi_k|X_k] = 0$. The other terms can be bounded with Assumption 1 and notation of $M$ (from Section 2) as follows:

$$\big\|b(X_k) + \delta_k\big\|^2 \leq 2M^2(1 + \|X_k\|^2),$$

$$\Big\|\big(\sigma(X_k) + \Delta_k\big)\epsilon_k(X_k)\Big\|^2 \leq 2M^2(1 + \|X_k\|^2), \tag{9}$$

$$\langle X_k, b(X_k) + \delta_k\rangle \leq \tfrac{1}{2}\big(\|X_k\|^2 + \|b(X_k) + \delta_k\|^2\big) \leq (1 + M^2)(1 + \|X_k\|^2).$$

It gives the next estimate:

$$\begin{aligned}
1 + \mathbb{E}\|X_{k+1}\|^2 \leq & 1 + \mathbb{E}\|X_k\|^2 + \eta^2\mathbb{E}\big\|b(X_k) + \delta_k\big\|^2 \\
& + \eta^{1+\gamma}\mathbb{E}\Big\|\big(\sigma(X_k) + \Delta_k\big)\epsilon_k(X_k)\Big\|^2 + 2\eta\mathbb{E}\langle X_k, b(X_k) + \delta_k\rangle \\
\leq & 1 + \mathbb{E}\|X_k\|^2 + 2\big(\eta^2 + \eta^{1+\gamma} + 2\eta\big)(1 + M^2)(1 + \mathbb{E}\|X_k\|^2) \\
\leq & \big(1 + \eta \cdot 2(1 + M^2)(2 + \eta^\gamma + \eta)\big)(1 + \mathbb{E}\|X_k\|^2),
\end{aligned}$$

Running the recursion with $C = 2(1 + M^2)(2 + \eta^\gamma + \eta)$, we obtain:

$$1 + \mathbb{E}\|X_k\|^2 \leq \big(1 + C\eta\big)^k(1 + \|x_0\|^2) = e^{k\log(1+C\eta)}(1 + \|x_0\|^2) \leq e^{Ck\eta}(1 + \|x_0\|^2).$$

Using that $\eta \leq 1$, we get that $C = 2(1 + M^2)(2 + \eta^\gamma + \eta) \leq 8(1 + M^2)$. It finishes the proof. $\square$

**Lemma 6.** *Let Assumption 1 holds. Then for any $k, i \in \mathbb{N}_0$, $S \in \mathbb{N}$ (such that $i < S$):*

$$\mathbb{E}\big[\|X_{Sk+i} - X_{Sk}\|^2\big|X_{Sk}\big] \leq C'i\eta\big(1 + \|X_{Sk}\|^2\big),$$

*where $C' = 12M^2e^{(C+1)S\eta}$ and $C$ from Lemma 5.*

*Proof.* Similarly to Lemma 5 we obtain:

$$\begin{aligned}
\|X_{Sk+i+1} - X_{Sk}\|^2 = & \|X_{Sk+i+1} - X_{Sk+i} + (X_{Sk+i} - X_{Sk})\| \\
= & \|X_{Sk+i} - X_{Sk}\|^2 + \eta^2\big\|b(X_{Sk+i}) + \delta_{Sk+i}\big\|^2 \\
& + \eta^{1+\gamma}\Big\|\big(\sigma(X_{Sk+i}) + \Delta_{Sk+i}\big)\epsilon_{Sk+i}(X_k)\Big\|^2
\end{aligned}$$

$$+ 2\eta\langle X_{Sk+i} - X_{Sk}, b(X_{Sk+i}) + \delta_{Sk+i}\rangle + \xi_{Sk+i},$$

where $\xi_{Sk+i} = 2\sqrt{\eta^{1+\gamma}}\langle(\sigma(X_{Sk+i}) + \Delta_{Sk+i})\epsilon_{Sk+i}(X_{Sk+i}), X_{Sk+i} + \eta(b(X_{Sk+i}) + \delta_{Sk+i})\rangle$.
With unbiasedness of $e_{Sk+i}(X_{Sk+i})$, we have that $\mathbb{E}[\xi_{Sk+i}|e_{Sk+i}(X_{Sk+i})] = 0$. Now, using the bound equation 9, taking the conditional expectation and applying Lemma 5 for $X_i' = X_{Sk+i}, X_0' = X_{Sk}$, we deduce:

$$
\begin{aligned}
\mathbb{E}\big[\|X_{Sk+i+1} - X_{Sk}\|^2\big|X_{Sk}\big] =& \mathbb{E}\big[\|X_{Sk+i} - X_{Sk}\|^2\big|X_{Sk}\big] + \eta^2\mathbb{E}\big[\|b(X_{Sk+i}) + \delta_{Sk+i}\|^2\big|X_{Sk}\big] \\
&+ \eta^{1+\gamma}\mathbb{E}\big[\|(\sigma(X_{Sk+i}) + \Delta_{Sk+i})\epsilon_{Sk+i}(X_k)\|^2\big|X_{Sk}\big] \\
&+ 2\eta\mathbb{E}\big[\langle X_{Sk+i} - X_{Sk}, b(X_{Sk+i}) + \delta_{Sk+i}\rangle + \mathbb{E}\big[\xi_{Sk+i}\big|X_{Sk}\big] \\
\leq& (1+\eta)\mathbb{E}\big[\|X_{Sk+i} - X_{Sk}\|^2 + (\eta + \eta^2)\|b(X_{Sk+i}) + \delta_{Sk+i}\|^2\big|X_{Sk}\big] \\
&+ \eta^{1+\gamma}\mathbb{E}\big[\|(\sigma(X_{Sk+i}) + \Delta_{Sk+i})\epsilon_{Sk+i}(X_{Sk+i})\|^2\big|X_{Sk}\big] \\
\leq& (1+\eta)\mathbb{E}\big[\|X_{Sk+i} - X_{Sk}\|^2\big|X_{Sk}\big] \\
&+ 2(\eta + \eta^{1+\gamma} + \eta^2)M^2\big(1 + \mathbb{E}\big[\|X_{Sk+i}\|^2\big|X_{Sk}\big]\big) \\
\leq& (1+\eta)\mathbb{E}\big[\|X_{Sk+i} - X_{Sk}\|^2\big|X_{Sk}\big] \\
&+ 2(\eta + \eta^{1+\gamma} + \eta^2)M^2e^{Ci\eta}\big(1 + \|X_{Sk}\|^2\big).
\end{aligned}
$$

Recursively expanding the bound we obtain:

$$
\begin{aligned}
\mathbb{E}\big[\|X_{Sk+i} - X_{Sk}\|^2\big|X_{Sk}\big] &\leq \sum_{j=0}^{i-1}(1+\eta)^j 2M^2e^{Ci\eta}(\eta + \eta^{1+\gamma} + \eta^2)\big(1 + \|X_{Sk}\|^2\big) \\
&\leq 2M^2(\eta + \eta^{1+\gamma} + \eta^2)e^{(C+1)i\eta}i\eta\big(1 + \|X_{Sk}\|^2\big).
\end{aligned}
$$

With $i < S$ and $\eta \leq 1$, we get

$$\mathbb{E}\big[\|X_{Sk+i} - X_{Sk}\|^2\big|X_{Sk}\big] \leq 12M^2e^{(C+1)S\eta}i\eta\big(1 + \|X_{Sk}\|^2\big).$$

$\square$

*Remark* 1. In Lemma 1 and after we use that $S = \eta^{-\frac{1-\beta}{2}}(1 - \chi_0) + \chi_0$, then $S\eta = \eta^{\frac{1+\beta}{2}}(1 - \chi_0) + \chi_0\eta \leq 1$, and the result of Lemma 6 can be rewritten immediately as

$$\mathbb{E}\big[\|X_{Sk+i} - X_{Sk}\|^2\big|X_{Sk}\big] \leq C'i\eta\big(1 + \|X_{Sk}\|^2\big),$$

where $C' = 12M^2e^{(C+1)}$ and $C = 8(1 + M^2)$.

## B.2 Proofs of Lemmas 1-3

**Lemma 7** (Lemma 1). *Let Assumption 1 holds. If $L \geq 1 + 10M_0^2$, $\overline{\eta}L \leq 1$ and $S\eta \leq 1$, then for any $t > 0$:*

$$\Delta_t^S = \max_{\overline{\eta}k' \leq t} \frac{\mathbb{E}\|X_{Sk'} - Y_{\overline{\eta}k'}^X\|^2}{R^2(t)} = C''\left(\eta^{2\alpha} + \frac{\eta^{\gamma+\beta}}{S}(1 - \chi_0) + \overline{\eta}\right),$$

*where $R^2(t) \geq \max\{1; \max_{\eta k \leq t}\mathbb{E}\|X_k\|^2\}$ and*
$C'' = 2\big(3M_1^2e^C + 3M_0^2C' + 4M_1^2e^C + 48M_0^2M^2e^{C+1} + 4d\sigma_1^2M_e^2\big)$ *where the constants $C, C'$ are defined in Lemmas 6 and 5) respectively.*

*Proof.* Let us consider the difference between $Y_{\overline{\eta}(k+1)}^X$ and $X_{S(k+1)}$, square it, take the expectation (conditional on $X_{Sk}$) and get:

$$
\begin{aligned}
\mathbb{E}&\big[\|Y_{\overline{\eta}(k+1)}^X - X_{S(k+1)}\|^2\big|X_{Sk}\big] \\
&= (1 - L\overline{\eta})^2\mathbb{E}\big[\|Y_{\overline{\eta}k}^X - X_{Sk}\|^2\big|X_{Sk}\big] + \frac{\overline{\eta}^2}{S^2}\mathbb{E}\Big[\Big\|\sum_{i=0}^{S-1}\big(b(X_{Sk+i}) - b(Y_{\overline{\eta}k}^X) + \delta_{Sk+i}\big)\Big\|^2\Big|X_{Sk}\Big]
\end{aligned}
$$

$$+ 2(1 - L\overline{\eta})\overline{\eta} \sum_{i=0}^{S-1} \mathbb{E}\big[\langle Y_{\overline{\eta}k}^X - X_{Sk}, \frac{1}{S}(b(X_{Sk+i}) - b(Y_{\overline{\eta}k}^X) + \delta_{Sk+i})\rangle \big| X_{Sk}\big]$$

$$+ \overline{\eta}\eta^\gamma \mathbb{E}\Big[\Big\| \frac{1}{\sqrt{S}} \sum_{i=0}^{S-1} (\sigma(X_{Sk+i}) + \Delta_{Sk+i})\epsilon_{Sk+i}(X_{Sk+i}) - \sigma(Y_{\overline{\eta}k}^X)\zeta_k^S(X_{Sk})\Big\|^2 \Big| X_{Sk}\Big]$$

$$\leq \big((1 - L\overline{\eta})^2 + \overline{\eta}(1 - L\overline{\eta})\big)\mathbb{E}\big[\|Y_{\overline{\eta}k}^X - X_{Sk}\|^2 \big| X_{Sk}\big]$$

$$+ \frac{\overline{\eta}^2 + \overline{\eta}(1 - L\overline{\eta})}{S^2} \mathbb{E}\Big[\Big\| \sum_{i=0}^{S-1} \big(b(X_{Sk+i}) - b(Y_{\overline{\eta}k}^X) + \delta_{Sk+i}\big)\Big\|^2 \Big| X_{Sk}\Big]$$

$$+ \overline{\eta}\eta^\gamma \mathbb{E}\Big[\Big\| \frac{1}{\sqrt{S}} \sum_{i=0}^{S-1} (\sigma(X_{Sk+i}) + \Delta_{Sk+i})\epsilon_{Sk+i}(X_{Sk+i}) - \sigma(Y_{\overline{\eta}k}^X)\zeta_k^S(X_{Sk})\Big\|^2 \Big| X_{Sk}\Big].$$
$$(10)$$

In the first step, we also used the unbiasedness of $\epsilon_{Sk+i}(X_{Sk+i})$ (in the same way as in Lemmas 5, 6), this makes all inner products with $\epsilon_{Sk+i}(X_{Sk+i})$ equal to 0 in the expectation. In the second step, we use that $\overline{\eta}L \leq 1$. We start with bounding the last term. One can make the following estimate with Assumption 1 and Lemmas 5, 6:

$$\mathbb{E}\Big[\Big\| \frac{1}{\sqrt{S}} \sum_{i=0}^{S-1}(\sigma(X_{Sk+i}) + \Delta_{Sk+i})\epsilon_{Sk+i}(X_{Sk+i}) - \sigma(Y_{\overline{\eta}k}^X)\zeta_k^S(X_{Sk})\Big\|^2 \Big| X_{Sk}\Big]$$

$$\leq 4\mathbb{E}\Big[\Big\| \frac{1}{\sqrt{S}} \sum_{i=0}^{S-1} \Delta_{Sk+i}\epsilon_{Sk+i}(X_{Sk+i})\Big\|^2 \Big| X_{Sk}\Big]$$

$$+$$

$$+ 4\mathbb{E}\Big[\Big\| \sigma(Y_{\overline{\eta}k}^X)\big(\frac{1}{\sqrt{S}} \sum_{i=0}^{S-1} \epsilon_{Sk+i}(X_{Sk}) - \zeta_k^S(X_{Sk})\big)\Big\|^2 \Big| X_{Sk}\Big]$$

$$+ 4\mathbb{E}\Big[\Big\| \big(\sigma(X_{Sk}) - \sigma(Y_{\overline{\eta}k}^X)\big)\zeta_k^S(X_{Sk})\Big\|^2 \Big| X_{Sk}\Big].$$

Next, we use that for $i < j$: $\mathbb{E}\big[\langle\epsilon_{Sk+i}(X_{Sk+i}), \epsilon_{Sk+j}(X_{Sk+j})\rangle \big| X_{Sk}\big] = \mathbb{E}\big[\langle\epsilon_{Sk+i}(X_{Sk+i}), \mathbb{E}_{\epsilon_{Sk+j}}[\epsilon_{Sk+j}(X_{Sk+j})]\rangle \big| X_{Sk}\big] = 0$, and get

$$\mathbb{E}\Big[\Big\| \frac{1}{\sqrt{S}} \sum_{i=0}^{S-1}(\sigma(X_{Sk+i}) + \Delta_{Sk+i})\epsilon_{Sk+i}(X_{Sk+i}) - \zeta_k^S(X_{Sk})\Big\|^2 \Big| X_{Sk}\Big]$$

$$\leq \frac{4}{S} \sum_{i=0}^{S-1} \mathbb{E}\Big[\Big\| \Delta_{Sk+i}\epsilon_{Sk+i}(X_{Sk+i})\Big\|^2 \Big| X_{Sk}\Big]$$

$$+ \frac{4}{S} \sum_{i=0}^{S-1} \mathbb{E}\Big[\Big\| \big(\sigma(X_{Sk+i})\epsilon_{Sk+i}(X_{Sk+i}) - \sigma(X_{Sk})\epsilon_{Sk+i}(X_{Sk})\big)\Big\|^2 \Big| X_{Sk}\Big]$$

$$+ 4\mathbb{E}\Big[\Big\| \sigma(X_{Sk})\big(\frac{1}{\sqrt{S}} \sum_{i=0}^{S-1} \epsilon_{Sk+i}(X_{Sk}) - \zeta_k^S(X_{Sk})\big)\Big\|^2 \Big| X_{Sk}\Big]$$

$$+ 4\mathbb{E}\Big[\Big\| \big(\sigma(X_{Sk}) - \sigma(Y_{\overline{\eta}k}^X)\big)\zeta_k^S(X_{Sk})\big)\Big\|^2 \Big| X_{Sk}\Big].$$

Assumption 1 gives

$$\mathbb{E}\Big[\Big\| \frac{1}{\sqrt{S}} \sum_{i=0}^{S-1}(\sigma(X_{Sk+i}) + \Delta_{Sk+i})\epsilon_{Sk+i}(X_{Sk+i}) - \sigma(Y_{\overline{\eta}k}^X)\zeta_k^S(X_{Sk})\Big\|^2 \Big| X_{Sk}\Big]$$

$$\leq \frac{4}{S} \sum_{i=0}^{S-1} M_1^2 \eta^{2\alpha-\gamma}\big(1 + \mathbb{E}\big[\big\| X_{Sk+i}\big\|^2 \big| X_{Sk}\big]\big)$$

$$+ \frac{4}{S} \sum_{i=0}^{S-1} M_0^2 \mathbb{E}\big[\|X_{Sk+i} - X_{Sk}\|^2 \big| X_{Sk}\big]$$

$$+ 4d\sigma_1^2 \mathbb{E}\Big[\Big\|\frac{1}{\sqrt{S}} \sum_{i=0}^{S-1} \epsilon_{Sk+i}(X_{Sk}) - \zeta_k^S(X_{Sk})\Big\|^2 \Big| X_{Sk}\Big]$$

$$+ 4M_0^2 \Big\|X_{Sk} - Y_{\bar{\eta}k}^X\Big\|^2,$$

where for the last term, we used the following chain of identifies derived from Assumption 1 and noting that $\mathbb{E}\langle \sigma(x)(\epsilon_k(x) - \epsilon_k(x')), \sigma(x)(\epsilon_k(x) + \epsilon_k(x'))\rangle = \mathbb{E}\langle \sigma(x')(\epsilon_k(x) - \epsilon_k(x')), \sigma(x')(\epsilon_k(x) + \epsilon_k(x'))\rangle = 0$, $\mathbb{E}\|(\sigma(x) + \sigma(x'))(\epsilon_k(x) - \epsilon_k(x'))\|^2 \geq \mathbb{E}\|(\sigma(x) - \sigma(x'))(\epsilon_k(x) - \epsilon_k(x'))\|^2$:

$$M_0^2 \|x - x'\|^2 \geq \mathbb{E}\|\sigma(x)\epsilon_k(x) - \sigma(x')\epsilon_k(x')\|^2$$

$$= \frac{1}{4}\mathbb{E}\Big\|\big(\sigma(x) + \sigma(x')\big)(\epsilon_k(x) - \epsilon_k(x')) + \big(\sigma(x) - \sigma(x')\big)(\epsilon_k(x) + \epsilon_k(x'))\Big\|^2$$

$$= \frac{1}{4}\mathbb{E}\Big\|\big(\sigma(x) + \sigma(x')\big)(\epsilon_k(x) - \epsilon_k(x'))\Big\|^2 + \frac{1}{4}\mathbb{E}\big\|\big(\sigma(x) - \sigma(x')\big)(\epsilon_k(x) + \epsilon_k(x'))\big\|^2$$

$$+ \frac{1}{2}\mathbb{E}\Big\langle \big(\sigma^2(x) - \sigma^2(x')\big)(\epsilon_k(x) - \epsilon_k(x')), \epsilon_k(x) - \epsilon_k(x')\Big\rangle$$

$$= \frac{1}{4}\mathbb{E}\Big\|\big(\sigma(x) + \sigma(x')\big)(\epsilon_k(x) - \epsilon_k(x'))\Big\|^2 + \frac{1}{4}\mathbb{E}\big\|\big(\sigma(x) - \sigma(x')\big)(\epsilon_k(x) + \epsilon_k(x'))\big\|^2$$

$$+ \frac{1}{2}\mathbb{E}\Big\langle \sigma(x)(\epsilon_k(x) - \epsilon_k(x')), \sigma(x)(\epsilon_k(x) - \epsilon_k(x'))\Big\rangle$$

$$- \frac{1}{2}\mathbb{E}\Big\langle \sigma(x')(\epsilon_k(x) - \epsilon_k(x')), \sigma(x)(\epsilon_k(x) - \epsilon_k(x'))\Big\rangle$$

$$= \frac{1}{4}\mathbb{E}\Big\|\big(\sigma(x) + \sigma(x')\big)(\epsilon_k(x) - \epsilon_k(x'))\Big\|^2 + \frac{1}{4}\mathbb{E}\big\|\big(\sigma(x) - \sigma(x')\big)(\epsilon_k(x) + \epsilon_k(x'))\big\|^2$$

$$\geq \frac{1}{4}\mathbb{E}\Big\|\big(\sigma(x) - \sigma(x')\big)(\epsilon_k(x) - \epsilon_k(x'))\Big\|^2 + \frac{1}{4}\mathbb{E}\big\|\big(\sigma(x) - \sigma(x')\big)(\epsilon_k(x) + \epsilon_k(x'))\big\|^2$$

$$\geq \frac{1}{2}\mathbb{E}\big\|(\sigma(x) - \sigma(x'))\epsilon_k(x)\big\|^2 + \frac{1}{2}\mathbb{E}\big\|(\sigma(x) - \sigma(x'))\epsilon_k(x')\big\|^2$$

$$\geq \|\sigma(x) - \sigma(x')\|_F^2$$

Taking into account that $\zeta_k^S(X_{Sk})$ is an optimal coupling (Section 3.1), we can estimate $\big\|\frac{1}{\sqrt{S}} \sum_{i=0}^{S-1} \epsilon_{Sk+i}(X_{Sk}) - \zeta_k^S(X_{Sk})\big\|^2$ by Assumption 1.

$$\mathbb{E}\Big[\Big\|\frac{1}{\sqrt{S}} \sum_{i=0}^{S-1} \big(\sigma(X_{Sk+i}) + \Delta_{Sk+i}\big)\epsilon_{Sk+i}(X_{Sk+i}) - \zeta_k^S(X_{Sk})\Big\|^2 \Big| X_{Sk}\Big]$$

$$\leq \frac{4}{S} \sum_{i=0}^{S-1} M_1^2 \eta^{2\alpha-\gamma}\big(1 + \mathbb{E}\big[\|X_{Sk+i}\|^2 \big| X_{Sk}\big]\big)$$

$$+ \frac{4}{S} \sum_{i=0}^{S-1} M_0^2 \mathbb{E}\big[\|X_{Sk+i} - X_{Sk}\|^2 \big| X_{Sk}\big]$$

$$+ \frac{4}{S} d\sigma_1^2 M_e^2 \eta^\beta + 4M_0^2 \Big\|X_{Sk} - Y_{\bar{\eta}k}^X\Big\|^2.$$

With Lemmas 5, 6, one can obtain

$$\mathbb{E}\Big[\Big\|\frac{1}{\sqrt{S}} \sum_{i=0}^{S-1} \big(\sigma(X_{Sk+i}) + \Delta_{Sk+i}\big)\epsilon_{Sk+i}(X_{Sk+i}) - \zeta_k^S(X_{Sk})\Big\|^2 \Big| X_{Sk}\Big]$$

$$\leq \frac{4}{S} \sum_{i=0}^{S-1} M_1^2 \eta^{2\alpha-\gamma} e^{Ci\eta}\big(1 + \|X_{Sk}\|^2\big) + \frac{4}{S} \sum_{i=0}^{S-1} M_0^2 \cdot 12M^2 e^{(C+1)S\eta} i\eta\big(1 + \|X_{Sk}\|^2\big)$$

$$+ \frac{4}{S} d\sigma_1^2 M_e^2 \eta^\beta + 4M_0^2 \mathbb{E}\Big[\Big\|X_{Sk} - Y_{\bar{\eta}k}^X\Big\|^2 \Big| X_{Sk}\Big]$$

$$\leq 4M_1^2\eta^{2\alpha-\gamma}e^{CS\eta}\big(1+\|X_{Sk}\|^2\big) + 48M_0^2M^2e^{(C+1)S\eta}S\eta\big(1+\|X_{Sk}\|^2\big)$$
$$+ \frac{4d\sigma_1^2M_e^2\eta^\beta}{S} + 4M_0^2\mathbb{E}\Big[\Big\|X_{Sk}-Y_{\overline\eta k}^X\Big\|^2\Big|X_{Sk}\Big]$$
$$\leq \Big(4M_1^2e^{CS\eta} + 48M_0^2M^2e^{(C+1)S\eta} + 4d\sigma_1^2M_e^2\Big)\Big(\eta^{2\alpha-\gamma}+S\eta+\frac{\eta^\beta(1-\chi_0)}{S}\Big)\big(1+\|X_{Sk}\|^2\big)$$
$$+ 4M_0^2\Big\|X_{Sk}-Y_{\overline\eta k}^X\Big\|^2.$$

Here we also used that $M_e = 0$ if $\chi_0 = 0$. Taking into account $S\eta \leq 1$, we can deal with $e^{CS\eta} \leq e^C$:

$$\mathbb{E}\Big[\Big\|\frac{1}{\sqrt S}\sum_{i=0}^{S-1}\big(\sigma(X_{Sk+i})+\Delta_{Sk+i}\big)\epsilon_{Sk+i}(X_{Sk+i})-\zeta_k^S(X_{Sk})\Big\|^2\Big|X_{Sk}\Big]$$
$$\leq \Big(4M_1^2e^C + 48M_0^2M^2e^{(C+1)} + 4d\sigma_1^2M_e^2\Big)\Big(\eta^{2\alpha-\gamma}+\overline\eta+\frac{\eta^\beta(1-\chi_0)}{S}\Big)\big(1+\|X_{Sk}\|^2\big)$$
$$+ 4M_0^2\Big\|X_{Sk}-Y_{\overline\eta k}^X\Big\|^2. \tag{11}$$

For the drift related terms we also use Assumption 1 and then Lemmas 5, 6:

$$\mathbb{E}\Big[\Big\|\sum_{i=0}^{S-1}\big(b(X_{Sk+i})-b(Y_{\overline\eta k}^X)+\delta_{Sk+i})\big)\Big\|^2\Big|X_{Sk}\Big]$$
$$\leq S\sum_{i=0}^{S-1}\mathbb{E}\Big[\Big\|b(X_{Sk+i})-b(Y_{\overline\eta k}^X)+\delta_{Sk+i})\Big\|^2\Big|X_{Sk}\Big]$$
$$\leq S\sum_{i=0}^{S-1}\mathbb{E}\Big[3\|\delta_{Sk+i}\|^2 + 3\|b(X_{Sk+i})-b(X_{Sk})\|^2 + 3\|b(X_{Sk})-b(Y_{\overline\eta k}^X)\|^2\Big|X_{Sk}\Big]$$
$$\leq S\sum_{i=0}^{S-1}\Big(3M_1^2\eta^{2\alpha}\big(1+\mathbb{E}\big[\big\|X_{Sk+i}\big\|^2\big|X_{Sk}\big]\big) + 3M_0^2\mathbb{E}\big[\|X_{Sk+i}-X_{Sk}\|^2\big|X_{Sk}\big]\Big)$$
$$+ 3S^2M_0^2\|X_{Sk}-Y_{\overline\eta k}^X\|^2$$
$$\leq S\sum_{i=0}^{S-1}\Big(3M_1^2\eta^{2\alpha}e^{Ci\eta}\big(1+\|X_{Sk}\|^2\big) + 3M_0^2C'i\eta\big(1+\|X_{Sk}\|^2\big)\Big)$$
$$+ 3S^2M_0^2\|X_{Sk}-Y_{\overline\eta k}^X\|^2$$
$$\leq S^2\big(3M_1^2\eta^{2\alpha}e^{CS\eta}+3M_0^2C'\overline\eta\big)\big(1+\|X_{Sk}\|^2\big) + 3S^2M_0^2\mathbb{E}\|Y_{\overline\eta k}^X-X_{Sk}\|^2.$$

Taking into account $S\eta \leq 1$, we can deal with $e^{CS\eta} \leq e^C$:

$$\mathbb{E}\Big[\Big\|\sum_{i=0}^{S-1}\big(b(X_{Sk+i})-b(Y_{\overline\eta k}^X)+\delta_{Sk+i})\big)\Big\|^2\Big|X_{Sk}\Big]$$
$$\leq S^2\big(3M_1^2\eta^{2\alpha}e^C+3M_0^2C'\overline\eta\big)\big(1+\|X_{Sk}\|^2\big) + 3S^2M_0^2\mathbb{E}\|Y_{\overline\eta k}^X-X_{Sk}\|^2. \tag{12}$$

Combining equation 10 with equation 11 and equation 12, we obtain:

$$\mathbb{E}\|Y_{\overline\eta(k+1)}^X-X_{S(k+1)}\|^2\big|X_{Sk}]$$
$$\leq \big(1-2L\overline\eta+L^2\overline\eta^2+\overline\eta-L\overline\eta^2\big)\|Y_{\overline\eta k}^X-X_{Sk}\|^2$$
$$+ (\overline\eta^2+\overline\eta(1-L\overline\eta))\big(3M_1^2\eta^{2\alpha}e^C+3M_0^2C'\overline\eta\big)\big(1+\|X_{Sk}\|^2\big)$$
$$+ (\overline\eta^2+\overline\eta(1-L\overline\eta))\cdot 3M_0^2\mathbb{E}\|Y_{\overline\eta k}^X-X_{Sk}\|^2$$
$$+ \overline\eta\eta^\gamma\cdot 4\big(M_1^2e^C+12M_0^2M^2e^{C+1}+d\sigma_1^2M_e^2\big)\Big(\eta^{2\alpha-\gamma}+\overline\eta+\frac{\eta^\beta(1-\chi_0)}{S}\Big)\big(1+\|X_{Sk}\|^2\big)$$
$$+ \overline\eta\eta^\gamma\cdot 4M_0^2\Big\|X_{Sk}-Y_{\overline\eta k}^X\Big\|^2$$

$$\leq \left(1 - \overline{\eta}(2L - L^2\overline{\eta} - 1 - 3M_0^2 - 3M_0^2\overline{\eta} - 4M_0^2\eta^\gamma)\right) \|Y_{\overline{\eta}k}^X - X_{Sk}\|^2$$
$$+ \overline{\eta} \left(3M_1^2 e^C + 3M_0^2 C' + 4M_1^2 e^C + 48M_0^2 M^2 e^{C+1} + 4d\sigma_1^2 M_e^2\right)$$
$$\cdot \left(\eta^{2\alpha} + \eta^{2\alpha}\overline{\eta} + \overline{\eta}^2 + \overline{\eta}^3 + \overline{\eta}\eta^\gamma + \frac{\eta^{\beta+\gamma}(1-\chi_0)}{S}\right) \left(1 + \|X_{Sk}\|^2\right).$$

With $L \geq 1 + 10M_0^2$, $L\overline{\eta} \leq 1$ and $\overline{\eta} = S\eta \leq 1$, we get

$$\mathbb{E}\|Y_{\overline{\eta}(k+1)}^X - X_{S(k+1)}\|^2 \big| X_{Sk}]$$
$$\leq 2 \left(3M_1^2 e^C + 3M_0^2 C' + 4M_1^2 e^C + 48M_0^2 M^2 e^{(C+1)} + 4d\sigma_1^2 M_e^2\right)$$
$$\cdot \left(\eta^{2\alpha} + \overline{\eta} + \frac{\eta^{\beta+\gamma}(1-\chi_0)}{S}\right) \left(1 + \|X_{Sk}\|^2\right).$$

Definition of $R^2(t)$ finishes the proof. $\qquad\square$

**Lemma 8** (Lemma 2). *Let Assumption 1 holds. Then for any $k \in \mathbb{N}_0$:*

$$\sup_{t \leq k\eta} \mathbb{E}\big\|Y_{[t/\overline{\eta}]\overline{\eta}}^X - Y_t\big\|^2 = C_2\overline{\eta}^{1+\gamma}R^2(k\eta),$$

*where $R^2(k\eta) \geq \max\left\{1; \max_{k' \leq k} \mathbb{E}\|X_{k'}\|^2\right\}$ and $C_2 \leq 4(L^2 + M^2)\left(1 + 3C''\right)\left(1 + \|x_0\|^2\right)$ (for the definitions of $L$ and $C''$ see Lemma 6).*

*Proof.* For $t = [t/\overline{\eta}]\overline{\eta} + \delta$ we write:

$$\mathbb{E}\big\|Y_{[t/\overline{\eta}]\overline{\eta}}^X - Y_t\big\|^2$$
$$\leq 2\delta^2 L^2 \mathbb{E}\big\|Y_{[t/\overline{\eta}]\overline{\eta}}^X - X_{[t/\overline{\eta}]S}\big\|^2 + \delta M^2(2\delta + 2\eta^\gamma)(1 + \mathbb{E}\big\|Y_{[t/\overline{\eta}]\overline{\eta}}^X\big\|^2)$$
$$\leq (2\delta^2 M^2 + 4M^2\eta^\gamma\delta)(1 + \mathbb{E}\big|X_{[t/\overline{\eta}]S}\big|^2)$$
$$+ (2\delta^2(L^2 + M^2) + 4M^2\eta^\gamma\delta)\mathbb{E}\big\|Y_{[t/\overline{\eta}]\overline{\eta}}^X - X_{[t/\overline{\eta}]S}\big\|^2$$
$$\leq 2M^2 S\eta\Big((\overline{\eta} + 2\eta^\gamma)(1 + \|x_0\|^2)e^{Ck\eta} + (S\eta(\tfrac{L^2}{M^2} + 1) + 2\eta^\gamma)\max_{\overline{\eta}k' \leq \eta k}\mathbb{E}\big\|Y_{\overline{\eta}k'}^X - X_{k'}\big\|^2\Big)$$
$$\leq \Big((2(L^2 + M^2)\overline{\eta}^2 + 2M^2\overline{\eta}^{1+\gamma}\Big)\Big((1 + \|x_0\|^2)e^{Ck\eta} + \max_{\overline{\eta}k' \leq \eta k}\mathbb{E}\big\|Y_{\overline{\eta}k'}^X - X_{k'}\big\|^2\Big).$$

$\qquad\square$

**Lemma 9** (Lemma 3). *Let Assumption 1 holds. Then for $L_1 \geq 2M_0 + 4M_0\eta^\gamma$ and any time horizon $t = k\eta \geq 0$:*

$$\mathcal{W}_2^2(\mathcal{L}(Y_t), \mathcal{L}(Z_t^Y)) \leq \mathbb{E}\big\|Y_t - Z_t^Y\big\|^2 \leq \sup_{t' \leq t} \mathbb{E}\big\|Y_{[t'/\overline{\eta}]\overline{\eta}}^X - Y_{t'}\big\|^2.$$

*Proof.* Consider process $Q_t = Y_t - Z_t^Y$. Note that $Q_0 = 0_d$. We have the following SDE for the process $Q_t$:

$$dQ_t = -L_1 Q_t dt + \left(b(Y_{[t/\overline{\eta}]\overline{\eta}}^X) - b(Z_t^Y)\right)dt + \sqrt{2\eta^\gamma}(\sigma(Y_{[t/\overline{\eta}]\overline{\eta}}^X) - \sigma(Z_t^Y))dW_t$$
$$= -L_1 Q_t dt + (b(Y_t) - b(Z_t^Y))dt + (b(Y_{[t/\overline{\eta}]\overline{\eta}}^X) - b(Y_t))dt$$
$$+ \sqrt{2\eta^\gamma}(\sigma(Y_t) - \sigma(Z_t^Y))dW_t + \sqrt{2\eta^\gamma}(\sigma(Y_{[t/\overline{\eta}]\overline{\eta}}^X) - \sigma(Y_t))dW_t.$$

Now, by applying Itô lemma (see Theorem 8.1, p. 220 and Remark 9.1 on p. 257 in Baldi & Baldi (2017)) to $f(x) = x^2$ we obtain the following bound on the process $\|Q_t\|^2$:

$$\|Q_t\|^2 \leq \int_0^t \Big(-2L_1\|Q_t\|^2 + 2\langle Q_t, b(Y_t) - b(Z_t^Y)\rangle + 4\eta^\gamma \text{tr}\left(\left(\sigma(Y_t) - \sigma(Z_t^Y)\right)^2\right)$$
$$+ 2\big\|b(Y_{[t/\overline{\eta}]\overline{\eta}}^X) - b(Y_t)\big\|^2 + 4\eta^\gamma\text{tr}\left(\left(\sigma(Y_{[t/\overline{\eta}]\overline{\eta}}^X) - \sigma(Y_t)\right)^2\right)\Big)dt + \xi_t$$

$$\leq \int_0^t \left( (-2L_1 + 2M_0 + 4M_0^2\eta^\gamma)\|Q_t\|^2 + (2M_0 + 4M_0^2\eta^\gamma)\left\|Y_{[t/\overline{\eta}]\overline{\eta}}^X - Y_t\right\|^2 \right) \mathrm{d}t + \xi_t,$$

where $\xi_t = \sqrt{2\eta^\gamma} \int_0^t \langle 2Q_t, (\sigma(Y_{[t/\overline{\eta}]\overline{\eta}}^X) - \sigma(Z_t^Y))\mathrm{d}W_t \rangle$.

Next, by noting that $\xi_0 = 0$ and $\xi_t$ by it's definition is a martingale process we have $\mathbb{E}\xi_t = 0$ (see Remark 9.2 on p. 264 in Baldi & Baldi (2017)) and, thus, by taking expectation $\mathbb{E}\|Q_t\|^2$ we arrive at the following bound:

$$\mathbb{E}\|Q_t\|^2 \leq \mathbb{E}\int_0^t \left( (-2L_1 + 2M_0 + 4M_0^2\eta^\gamma)\|Q_t\|^2 + (2M_0 + 4M_0^2\eta^\gamma)\left\|Y_{[t/\overline{\eta}]\overline{\eta}}^X - Y_t\right\|^2 \right) \mathrm{d}t$$

$$= \int_0^t \left( (-2L_1 + 2M_0 + 4M_0^2\eta^\gamma)\mathbb{E}\|Q_t\|^2 + (2M_0 + 4M_0^2\eta^\gamma)\mathbb{E}\left\|Y_{[t/\overline{\eta}]\overline{\eta}}^X - Y_t\right\|^2 \right) \mathrm{d}t$$

$$\leq \int_0^t \left( (-2L_1 + 2M_0 + 4M_0^2\eta^\gamma)\mathbb{E}\|Q_t\|^2 + (2M_0 + 4M_0^2\eta^\gamma)\sup_{t'\leq t}\mathbb{E}\left\|Y_{[t'/\overline{\eta}]\overline{\eta}}^X - Y_{t'}\right\|^2 \right) \mathrm{d}t$$

Recall that we can choose $L_1$ arbitrary in the definition of $Z_t^Y$. We want to make sure that $-2L_1 + 2M_0 + 4M_0^2\eta^\gamma < 0$. In that case, we will obtain that $\mathbb{E}\|Q_t\|^2 \leq \frac{(2M_0 + 4M_0^2\eta^\gamma)}{2L_1 - 2M_0 - 4M_0^2\eta^\gamma}\sup_{t'\leq t}\mathbb{E}\left\|Y_{[t'/\overline{\eta}]\overline{\eta}}^X - Y_{t'}\right\|^2$. To simplify the bound, we are going to require $\frac{(2M_0 + 4M_0^2\eta^\gamma)}{2L_1 - 2M_0 - 4M_0^2\eta^\gamma} \leq 1$. Thus, by considering $L_1$ such that $-2L_1 + 2M_0 + 4M_0^2\eta^\gamma \leq -(2M_0 + 4M_0^2\eta^\gamma)$ the lemma implies due to the fact that $\mathcal{W}_2^2(\mathcal{L}(Y_t), \mathcal{L}(Z_t^Y)) \leq \mathbb{E}\left\|Y_t - Z_t^Y\right\|^2 = \mathbb{E}\|Q_t\|^2$. $\square$

### B.3 PROOF OF THEOREM 1

The proof idea relies on multiple applications of Theorem 4 to get a process for which the classic Girsanov theorem 3 holds. To apply the theorem 3, we must ensure that the final process is Markovian diffusion, for which the Novikov type condition $\mathbb{E}e^{\int_0^T \left\|\overline{g}(\overline{Z}_t, t)\right\|^2 \mathrm{d}s} < \infty$ holds. While 4 will give us Markovian property, to get the Novikov condition, we are going to build a sequence of intermediate processes using stopping time $\tau_r = T \wedge \inf_{\tau\geq 0}\{\int_0^\tau \left\|\overline{g}(Z_t, t)\right\|^2 \mathrm{d}t \geq r^2\}$ and then we are going to select a subsequence $r_n \to \infty$ which in the limit will upper bound $D_{\mathrm{KL}}$ for the original process $\overline{Z}_t$. To do the former, we will prove the following Lemma first.

**Lemma 10.** *Assume that $Z_1, \ldots, Z_k, \ldots$ is a sequence of random variables converging in law to the random variable $Z_0$ and such that $\exists \xi_i \triangleq \frac{\mathrm{d}\mathcal{L}(Z_i)}{\mathrm{d}\mu}$ for some measure $\mu$ $\forall i \in \{0, 1, \ldots\}$. Moreover we assume that $D_{\mathrm{KL}}(\xi_i\mu\|\mu) < \infty \forall i \in \{0, 1, \ldots\}$ uniformly and that measure $\mu$ is non-singular with respect to standard Lebesgue measure on $\mathbb{R}^d$. Then it holds:*

$$D_{\mathrm{KL}}(\xi_0\mu\|\mu) \leq \sup_{k\geq 1} D_{\mathrm{KL}}(\xi_k\mu\|\mu) < \infty$$

*Proof.* Let's define the following unity partition system $\mathcal{P}_n = \left\{\xi_0^{-1}([m2^{-n}, (m+1)2^{-n})|0 \leq m \leq n2^n, m \in \mathbb{N}_0\right\} \cup \left\{\xi_0^{-1}([n, \infty))\right\}$. For each density $\xi_k$ we define

$$\Gamma_n(\xi_k) = \sum_{A\in\mathcal{P}_n} \mathbf{1}_A \frac{\mathbb{E}_\mu(\xi_k\mathbf{1}_A)}{\mu(A)} \geq 0$$

$\Gamma_m(\xi_k)$ is essentially finite-sum approximation of Lebesgue integral of $\xi_k$. Let $\xi_k^{<n} = \xi_k \wedge n = \min\{\xi_k, n\}$. We have:

$$\Gamma_n(\xi_k^{<n}) = \sum_{A\in\mathcal{P}_n} \mathbf{1}_A \frac{\mathbb{E}_\mu(\xi_n^{<n}\mathbf{1}_A)}{\mu(A)} = \sum_{A\in\mathcal{P}_n} \mathbf{1}_A \frac{\mathbb{E}_\mu(\xi_0^{<n}\mathbf{1}_A)}{\mu(A)} + \sum_{A\in\mathcal{P}_n} \mathbf{1}_A \frac{\mathbb{E}_\mu((\xi_n^{<n} - \xi_0^{<n})\mathbf{1}_A)}{\mu(A)}$$

Consider taking subsequence $k_n$ such that $\left|\mathbb{E}_\mu((\xi_{k_n}^{<n} - \xi_0^{<n})\mathbf{1}_A)\right| \leq 2^{-n}\mu(A)$. Since $\xi_k$ is convergent in law to $\xi_0$ we have that $\xi_k^{<n}$ convergent weakly to $\xi_0^{<n}$ as $k \to \infty$, hence for each $n$ and for each $A \in$

$\mathcal{P}_n$ there exists $k_{n,A}$ large enough so that $\forall k \geq k_{n,A}$ we have $\left|\mathbb{E}_\mu\big((\xi_{k_{n,A}}^{<n} - \xi_0^{<n})\mathbf{1}_A\big)\right| \leq 2^{-n}\mu(A)$. Since $\mathcal{P}_n$ is finite we can define $k_n = \max_{A \in \mathcal{P}_n} k_{n,A} < \infty$, which satisfies $\left|\mathbb{E}_\mu\big((\xi_{k_n}^{<n} - \xi_0^{<n})\mathbf{1}_A\big)\right| \leq 2^{-n}\mu(A)$ for all $A \in \mathcal{P}_n$.

Thus, we have:

$$\left|\Gamma_n(\xi_{k_n}^{<n}) - \sum_{A \in \mathcal{P}_n} \mathbf{1}_A \frac{\mathbb{E}_\mu\big(\xi_0^{<n}\mathbf{1}_A\big)}{\mu(A)}\right| \leq 2^{-n}$$

Moreover, observe that $\xi_0^{<n} \equiv \sum_{A \in \mathcal{P}_n} \xi_0^{<n}\mathbf{1}_A$ since $\mathcal{P}_n$ is unity partition. We have almost surely:

$$\left|\xi_0^{<n} - \sum_{A \in \mathcal{P}_n} \mathbf{1}_A \frac{\mathbb{E}_\mu\big(\xi_0^{<n}\mathbf{1}_A\big)}{\mu(A)}\right| \leq \sum_{A \in \mathcal{P}_n} \mathbf{1}_A \mathbb{E}_\mu\big((\operatorname{ess\,sup}_A \xi_0^{<n} - \operatorname{ess\,inf}_A \xi_0^{<n})\mathbf{1}_A\big) \leq \max_{A \in \mathcal{P}_n} \Delta_A \xi_0^{<n},$$

where $\Delta_A \xi_0^{<n} = \operatorname{ess\,sup}_A \xi_0^{<n} - \operatorname{ess\,inf}_A \xi_0^{<n}$. By definition of $\mathcal{P}_n$ we have that $\Delta_A \xi_0^{<n} \leq 2^{-n}$, which by combining both inequalities implies almost surely:

$$\xi_0^{<n} - 2^{-n+1} \leq \Gamma_n(\xi_{k_n}) \leq \xi_0^{<n} + 2^{-n+1} \leq \Gamma_n(\xi_{k_n}) + 2^{-n+2}$$

Note that $\Gamma_n(\mathbf{1}^{<n}) = \mathbf{1}$, therefore, by dominated convergence theorem Edmonds (1977) with Data Processing inequality 2 $D_{\mathrm{KL}}\big(\Gamma_n(\xi_{k_n})\mu\big|\big|\Gamma_n(\mathbf{1})\mu\big) \leq D_{\mathrm{KL}}\big(\xi_{k_n}\mu\big|\big|\mu\big)$ we obtain $\sup_{k \geq 1} D_{\mathrm{KL}}\big(\xi_k\mu\big|\big|\mu\big) \geq \lim_{n \to \infty} D_{\mathrm{KL}}\big(\Gamma_n(\xi_{k_n})\mu\big|\big|\mu\big) = D_{\mathrm{KL}}\big(\xi_0\mu\big|\big|\mu\big)$ which finishes the proof.

$\square$

*Proof of Theorem 1.* By applying Theorem 4 we obtain the process $(\overline{Z}_t)_{t \geq 0}$ that has the same one-time marginals as the process $(Z_t)_{t \geq 0}$:

$$\mathrm{d}\overline{Z}_t = (b(\overline{Z}_t) + \mathbb{E}\big(g_t^*\big|Z_t = \overline{Z}_t\big))\mathrm{d}t + \sigma(\overline{Z}_t)\mathrm{d}W_t$$

Denote by $\overline{g}(x,t) \triangleq \mathbb{E}\big(g_t^*\big|Z_t = x\big)$. We have that $D_{\mathrm{KL}}\big(\mathcal{L}(Z_T)\big|\big|\mathcal{L}(Z_T^*)\big) = D_{\mathrm{KL}}\big(\mathcal{L}(\overline{Z}_T)\big|\big|\mathcal{L}(Z_T^*)\big)$. Consider defining processes $Z_t^r$ for $r \geq 0$ as:

$$\mathrm{d}Z_t^r = (b(Z_t^r) + \overline{g}_r(Z_t^r, t))\mathrm{d}t + \sigma(Z_t^r)\mathrm{d}W_t$$

where we define the following progressively measurable process $\overline{g}_r(Z_t^r, t) = \mathbf{1}_{\{\int_0^t \left\|\overline{g}(Z_t^r,t)\right\|^2 < r^2\}} \overline{g}(Z_t^r, t)$. Denote by $\tau_r = T \wedge \inf_{\tau \geq 0}\{\int_0^\tau \left\|\overline{g}(\overline{Z}_t, t)\right\|^2 \mathrm{d}t \geq r^2\}$. Clearly that $\tau_r$ is a stopping time, i.e. $\{\tau_r = t\}$ is $\overline{Z}_t$-measurable $\forall t \geq 0$. Moreover, we have the following property $\overline{Z}_t \mathbf{1}_{t < \tau_r} = Z_t^r \mathbf{1}_{t < \tau_r}$ holding almost surely. Moreover, $\tau_r \to T$ as $r \to \infty$ due to the continuity of probability which implies that $Z_t^r \to \overline{Z}_t \forall t \in [0, T]$ holding almost surely as $r \to \infty$. And, finally, we can rewrite $\overline{g}_r(Z_t^r, t) = \mathbf{1}_{\{\int_0^t \left\|\overline{g}(Z_t^r,t)\right\|^2 < r^2\}} \overline{g}(Z_t^r, t) = \mathbf{1}_{\{t < \tau_r\}} \overline{g}(Z_t^r, t) = \mathbf{1}_{\{t < \tau_r\}} \overline{g}(\overline{Z}_t, t)$.

By applying Theorem 4 again to the process $(Z_t^r)_{t \geq 0}$ we obtain the process $(\overline{Z}_t^r)_{t \geq 0}$ that has the same one-time marginals:

$$\mathrm{d}\overline{Z}_t^r = (b(\overline{Z}_t^r) + \mathbb{E}\big(\overline{g}_r(Z_t^r, t)\big|Z_t^r = \overline{Z}_t^r\big))\mathrm{d}t + \sigma(\overline{Z}_t^r)\mathrm{d}W_t$$

Denote by $\widetilde{g}_r(x,t) = \mathbb{E}\big(\overline{g}_r(Z_t^r, t)\big|Z_t^r = x\big)$ and by $W_t^r \triangleq W_t + \int_0^t \sigma(\overline{Z}_s^r)^{-1}\widetilde{g}_r(\overline{Z}_t^r, t))\mathrm{d}s$. Consider process $Z_t^*$ in the probability space where $W_t^*$ is a Wiener process and satisfies:

$$\mathrm{d}Z_t^* = b(Z_t^*)\mathrm{d}t + \sigma(Z_t^*)\mathrm{d}W_t^*$$

Let $\mu_T \triangleq \mathcal{L}(Z_t^* : 0 \leq t \leq T)$ which is independent from $r$. Let $\nu_T^r \triangleq \mathcal{L}(\overline{Z}_t^r : 0 \leq t \leq T)$. Note that the Novikov condition holds $\mathbb{E}e^{\int_0^T \left\|\sigma(\overline{Z}_s^r)^{-1}\widetilde{g}_r(\overline{Z}_t^r,t)\right\|^2 \mathrm{d}s} \leq e^{\sigma_0^{-2}r^2T} < \infty \forall T > 0$. Then by applying the Theorem 3 to $\overline{Z}_T^r$ and $Z_T^*$ we obtain the following sequence of inequalities:

$$D_{\mathrm{KL}}\big(\nu_T^r\big|\big|\mu_T\big) \stackrel{\text{Girsanov, 1}}{=} \mathbb{E}\int_0^T \left\|\sigma(\overline{Z}_t^r)^{-1}\widetilde{g}_r(\overline{Z}_t^r,t)\right\|^2 \mathrm{d}t \stackrel{\text{Operator norm}}{\leq} \sigma_0^{-2}\int_0^T \mathbb{E}\left\|\widetilde{g}_r(\overline{Z}_t^r,t)\right\|^2 \mathrm{d}t$$

$$\stackrel{\text{Same marginals, 4}}{=} \sigma_0^{-2}\int_0^T \mathbb{E}\left\|\widetilde{g}_r(Z_t^r,t)\right\|^2 \mathrm{d}t \stackrel{\text{Jensen,}\|\mathbb{E}\cdot\|^2 \leq \mathbb{E}\|\cdot\|^2, 4}{\leq} \sigma_0^{-2}\int_0^T \mathbb{E}\left\|\overline{g}_r(Z_t^r,t)\right\|^2 \mathrm{d}t$$

$$\overset{\text{Due definition of }\tau_r,\,\overline{g}_r}{=} \sigma_0^{-2} \int_0^{\tau_r} \mathbb{E}\big\|\overline{g}_r(Z_t^r,t)\big\|^2 \mathrm{d}t \overset{\text{Due definition of }\tau_r,\,Z_t^r}{=} \sigma_0^{-2} \int_0^{\tau_r} \mathbb{E}\big\|\overline{g}(\overline{Z}_t,t)\big\|^2 \mathrm{d}t$$

$$\overset{\tau_r \leq T \text{ and } \|\cdot\|^2 \geq 0}{\leq} \sigma_0^{-2} \int_0^T \mathbb{E}\big\|\overline{g}(\overline{Z}_t,t)\big\|^2 \mathrm{d}t \overset{\text{Same marginals, 4}}{=} \sigma_0^{-2} \int_0^T \mathbb{E}\big\|\overline{g}(Z_t,t)\big\|^2 \mathrm{d}t$$

$$\overset{\text{Jensen},\,\|\mathbb{E}\cdot\|^2 \leq \mathbb{E}\|\cdot\|^2,\,4}{\leq} \sigma_0^{-2} \int_0^T \mathbb{E}\big\|g_t^*\big\|^2 \mathrm{d}t \overset{\text{By assumption}}{<} \infty,$$

where we use Jensen inequality as $\|\mathbb{E}\cdot\|^2 \leq \mathbb{E}\|\cdot\|^2$ whenever expression inside of $\|\cdot\|^2$ was obtained from application of Theorem 4 and use property of the same Theorem 4 that one-time marginals coincide for processes that we obtain from applying it, which implies equality of expectations.

Then we have that $\forall r \geq 0$ the following uniform majorization condition holds $D_{\mathrm{KL}}\big(\nu_T^r\big\|\mu_T\big) \leq \sigma_0^{-2} \int_0^T \mathbb{E}\big\|g_s^*\big\|^2 \mathrm{d}s < \infty$. By Data Processing inequality 2 we have $D_{\mathrm{KL}}\big(\mathcal{L}(\overline{Z}_T^r)\big\|\mathcal{L}(Z_T^*)\big) \leq D_{\mathrm{KL}}\big(\nu_T^r\big\|\mu_T\big)$. Using the fact that $\mathcal{L}(\overline{Z}_T^r) = \mathcal{L}(Z_T^r)$ we obtain majorization condition:

$$D_{\mathrm{KL}}\big(\mathcal{L}(Z_T^r)\big\|\mathcal{L}(Z_T^*)\big) \leq D_{\mathrm{KL}}\big(\nu_T^r\big\|\mu_T\big) \leq \sigma_0^{-2} \int_0^T \mathbb{E}\big\|g_s^*\big\|^2 \mathrm{d}s < \infty$$

By noting $\mathcal{L}(Z_T^r) \Rightarrow \mathcal{L}(\overline{Z}_T)$ as $r \to \infty$ and $D_{\mathrm{KL}}\big(\mathcal{L}(\overline{Z}_T)\big\|\mathcal{L}(Z_T^*)\big) < \infty$ by Lemma 10 we obtain:

$$D_{\mathrm{KL}}\big(\mathcal{L}(\overline{Z}_T)\big\|\mathcal{L}(Z_T^*)\big) \leq \sup_{r \geq 0} D_{\mathrm{KL}}\big(\mathcal{L}(Z_T^r)\big\|\mathcal{L}(Z_T^*)\big) \leq \sigma_0^{-2} \int_0^T \mathbb{E}\big\|g_s^*\big\|^2 \mathrm{d}s < \infty$$

Finally, using property $\mathcal{L}(Z_T) = \mathcal{L}(\overline{Z}_T)$ we obtain the desired:

$$D_{\mathrm{KL}}\big(\mathcal{L}(Z_T)\big\|\mathcal{L}(Z_T^*)\big) \leq \sigma_0^{-2} \int_0^T \mathbb{E}\big\|g_s^*\big\|^2 \mathrm{d}s$$

$\square$

### B.4 Proof of Lemma 4

*Proof.* Recall

$$\mathcal{C}_{\mathcal{W}}^2\big(\mathcal{L}(Z_t)\big) = \frac{1}{4} \min_{a > 0,\, z \in \mathbb{R}^d} \log \mathbb{E}^{\frac{1}{a}} \exp\Big(\frac{3}{2} + a\big\|Z_t - z\big\|^2\Big)$$

To prove that bound we consider the following ODE with $z(0) = x_0$:

$$\frac{\mathrm{d}z(t)}{\mathrm{d}t} = b(z(t))$$

Using it as $z \leftarrow z(t)$ and $a \leftarrow a(t) = \frac{M_0 \varepsilon}{4 d \sigma_1^2 \eta^\gamma} e^{-2mt}$ for $m \geq M_0(1 + \varepsilon)$ we consider process $S_t = \varphi(t, Z_t)$ for $\varphi(t, z) = e^{a(t)\|z - z(t)\|^2}$. Note that

$$\partial_t \varphi = a(t)\big(-2\langle b(z_t), z - z(t)\rangle - 2m\|z - z(t)\|^2\big)\varphi(t, z).$$

By applying Itô lemma (see Theorem 8.1, p. 220 and Remark 9.1 on p. 257 in Baldi & Baldi (2017)) to $S_t = \varphi(t, Z_t)$ and taking expectation, we obtain the equation:

$$\mathrm{d}\mathbb{E}S_t = \mathbb{E}\Big(\partial_t \varphi(t, Z_t) + \mathcal{A}\varphi(t, Z_t)\Big)\mathrm{d}t,$$

where we used the fact that $\mathbb{E}\big(\mathrm{d}W_t\big|W_t\big) = 0$ to eliminate all terms with $\mathrm{d}W_t$ (see Remark 9.2 on p. 264 in Baldi & Baldi (2017)). Substituting there definition of $\varphi$ we obtain:

$$\mathrm{d}\mathbb{E}\varphi(t, Z_t) = \mathbb{E}\Big(a(t)\big(-2\langle b(z_t), Z_t - z(t)\rangle - 2m\|z - z(t)\|^2\big)\varphi(t, z) + 2a(t)\langle b(z_t), Z_t - z(t)\rangle\varphi(t, z) +$$

$$+ \eta^\gamma \mathrm{Tr}\Big(\sigma(Z_t)\big(2a(t)I_d + 4a^2(t)(Z_t - z_t)(Z_t - z_t)^T\big)\sigma^T(Z_t)\Big)\varphi(t, Z_t)\Big)\mathrm{d}t$$

Simplifying yields:

$$\mathrm{d}\mathbb{E}\varphi(t, Z_t) = \mathbb{E}\Big(a(t)\big(-2m\|z - z(t)\|^2\big)\varphi(t, z) +$$

$$+\eta^\gamma \mathrm{Tr}\Big(\sigma(Z_t)\big(2a(t)I_d + 4a^2(t)(Z_t - z_t)(Z_t - z_t)^T\big)\sigma^T(Z_t)\Big)\varphi(t, Z_t)\Big)\mathrm{d}t$$

Since $\sigma(x) \le \sigma_1 I_d$ we can bound the last term as $4\eta^\gamma a^2(t)d\sigma_1^2\|Z_t - z_t\|^2$. Thus, by choosing $m \ge M_0(1+\varepsilon)$ we can ensure that $2a(t)m \ge 4\eta^\gamma a^2(t)d\sigma_1^2\|Z_t - z_t\|^2$ and, therefore, to upper bound all quadratic terms by zero, leaving contribution only from term $2a(t)\eta^\gamma \mathrm{Tr}\Big(\sigma(Z_t)\sigma^T(Z_t)\Big)\varphi(t, Z_t) \le 2a(t)\eta^\gamma d\sigma_1^2\varphi(t, Z_t)$. Which in turn allows us to reduce the estimate of $w(t) = \mathbb{E}e^{a(t)\|Z_t - z(t)\|^2}$ to deterministic Bihari-LaSalle inequality 5

$$w(t) \le \int_0^t w(t)\mathbf{d}A(t) + 1,$$

where we denoted as $A(t) = \big(\frac{\sigma_1^2 M_0 \varepsilon}{m}(1 - e^{-2mt})\big), A(0) = 0, A(t) \nearrow \frac{\sigma_1^2 M_0 \varepsilon}{m}$ as $t \to +\infty$. Using it gives us bound

$$w(t) \le \exp\Big(\big(\frac{\sigma_1^2 M_0 \varepsilon}{m}(1 - e^{-2mt})\big)\Big) \le \exp\big(\frac{\sigma_1^2 M_0 \varepsilon}{m}\big).$$

Finally, we obtain $\mathcal{C}_\mathcal{W}^2\big(\mathcal{L}(Z_t)\big) \le \eta^\gamma \frac{d\sigma_1^2}{M_0}e^{M_0(1+\varepsilon)t}\big(\frac{3}{2\varepsilon} + \frac{1}{1+\varepsilon}\big)$ by noting that $\mathcal{C}_\mathcal{W}^2\big(\mathcal{L}(Z_t)\big) \le \log\big(e^{\frac{3}{2}}w(t)\big)^{\frac{1}{a(t)}} = \frac{\frac{3}{2} + \log w(t)}{a(t)}$ and substituting formulas for $w(t)$ and $a(t)$, which concludes the proof.

$\square$

*Remark* 2. Constant $\varepsilon > 0$ can be chosen arbitrarily. smaller values lead to tighter asymptotic behavior, while larger values lead to tighter constants on finite horizon $T$. For simplicity, we choose $\varepsilon = 1$. Therefore, we have that

$$\mathcal{C}_\mathcal{W}\big(\mathcal{L}(Z_t)\big) \le \eta^{\frac{\gamma}{2}} \sigma_1 e^{M_0 t}\Big(\frac{2d}{M_0}\Big)^{\frac{1}{2}}$$

*Remark* 3. This result is aligned up to constant multipliers with the result of Remark 10.4 on p. 319 in Baldi & Baldi (2017). However, there is no explicit expression for some constants or proof of it in Baldi & Baldi (2017). Nevertheless, they claim that $a = a(t)$ can be selected arbitrarily as long as $a < \frac{e^{-2M_0 T}}{2T\sigma_1^2\eta^\gamma}$. This is slightly better asymotically for $T \to \infty$ as $\frac{1}{T} \ll e^{\varepsilon T}$ (as in our result) for arbitrarily small $\varepsilon > 0$. What's important here is that the order of growth $e^{\mathcal{O}(T)}$ is the same, and the scaling factor $\mathcal{O}(\frac{1}{\eta^\gamma})$ which is extremely important for the main result of our work, without which we won't be able to cover the case of $\gamma = 1$. While it may be interesting to derive the sharpest bound regarding exponential growth, we assume that horizon $T$ is fixed in our work. Therefore, the bound and order of the bound in Lemma 4 are sharp up to constant multipliers.

Moreover, in Baldi & Baldi (2017), this similar result is claimed to be true under the assumption of uniform ellipticity and uniformly bounded diffusion coefficient (i.e., $0 < \sigma_0 \le \sigma(x) \le \sigma_1 < \infty \forall x$) which is aligned with the Assumption 1.

### B.5 PROOF OF THEOREM 2

To prove the Theorem, we first will apply Theorem 1 to obtain the bound for the very last step in upper bounding, i.e., between $Z_t^Y$ and $Z_t$. Since that theorem gives bound on $D_{\mathrm{KL}}$ we are going to use transportation bound (Eq. 6) to obtain the bound on $\mathcal{W}_2$. After that, we have bounds between all subsequent pairs of processes that we have built: $X_{k'}, X_{Sk}, Y_{\overline{\eta}k}^X, Y_{\eta k'}, Z_{\eta k'}^Y, Z_{k'\eta}$. Thus, triangle inequality allows us to upper bound $\mathcal{W}_2\big(\mathcal{L}(X_{k'}), \mathcal{L}(Z_{k'\eta})\big)$ by the sum of bounds between each subsequent process.

**Corollary 6.** *We have that for time horizon $t = \eta k \ge 0$ the following bound holds:*

$$D_{\mathrm{KL}}\big(\mathcal{L}(Z_{k\eta}^Y)\big|\big|\mathcal{L}(Z_{k\eta})\big) \le \eta^{-\gamma}C_3 k\eta R^2(k\eta)\left(\eta^{2\alpha} + \frac{\eta^{\gamma+\beta}}{S}(1 - \chi_0) + \overline{\eta}\right),$$

*where $C_3 \le \frac{2(L^2 C'' + L_1^2 C_2)}{\sigma_0^2}$ (see Lemma 7 for the definition of $C''$, Lemma 8 for the definitions of $L$ and $C_2$, and Lemma 9 for the definition of $L$).*

*Proof.* Recall that $Z_t^Y$ has the following SDE:

$$\mathrm{d}Z_t^Y = \left(b(Z_t^Y) + G_t^S\right)\mathrm{d}t + \sqrt{\eta^\gamma}\sigma(Z_t^Y)\mathrm{d}W_t,$$

where $G_t^S = L(X_{[t/\overline{\eta}]S} - Y_{[t/\overline{\eta}]\overline{\eta}}^X) - L_1(Z_t^Y - Y_t)$. By applying Theorem 1 to $Z_t^Y$ and $Z_t$, we obtain the following:

$$D_{\mathrm{KL}}\left(\mathcal{L}(Z_{k\eta}^Y)\middle\|\mathcal{L}(Z_{k\eta})\right) \leq \frac{2}{\eta^\gamma {\sigma_0}^2}\int_0^{k\eta}\mathbb{E}\left\|G_t^S\right\|^2\mathrm{d}t$$

$$\leq \frac{2}{\eta^\gamma {\sigma_0}^2}\int_0^{k\eta}\mathbb{E}\left\|L(X_{[t/\overline{\eta}]S} - Y_{[t/\overline{\eta}]\overline{\eta}}^X) - L_1(Z_t^Y - Y_t)\right\|^2\mathrm{d}t$$

$$\leq \frac{2}{\eta^\gamma {\sigma_0}^2}\int_0^{k\eta}\left(L^2\mathbb{E}\left\|Y_{[t/\overline{\eta}]\overline{\eta}}^X - X_{[t/\overline{\eta}]S}\right\|^2 + L_1^2\mathbb{E}\left\|Z_t^Y - Y_t\right\|^2\right)\mathrm{d}t$$

By using Lemma 3 we obtain the bound:

$$D_{\mathrm{KL}}\left(\mathcal{L}(Z_{k\eta}^Y)\middle\|\mathcal{L}(Z_{k\eta})\right) \leq$$

$$\leq \frac{2}{\eta^\gamma {\sigma_0}^2}\int_0^{k\eta}\left(L^2\mathbb{E}\left\|Y_{[t/\overline{\eta}]\overline{\eta}}^X - X_{[t/\overline{\eta}]S}\right\|^2 + L_1^2\mathrm{sup}_{s\leq k\eta}\mathbb{E}\left\|Y_{[s/\overline{\eta}]\overline{\eta}}^X - Y_s\right\|^2\right)\mathrm{d}t$$

$$\leq \frac{2k\eta}{\eta^\gamma {\sigma_0}^2}\left(L^2\mathrm{sup}_{t\leq k\eta}\mathbb{E}\left\|Y_{[t/\overline{\eta}]\overline{\eta}}^X - X_{[t/\overline{\eta}]S}\right\|^2 + L_1^2\mathrm{sup}_{t\leq k\eta}\mathbb{E}\left\|Y_{[s/\overline{\eta}]\overline{\eta}}^X - Y_s\right\|^2\right)$$

Next, by using Lemmas 1 and 2 we bound both terms to obtain:

$$D_{\mathrm{KL}}\left(\mathcal{L}(Z_{k\eta}^Y)\middle\|\mathcal{L}(Z_{k\eta})\right) \leq \frac{2k\eta}{\eta^\gamma {\sigma_0}^2}\left(L^2 R^2(k\eta)C''\left(\eta^{2\alpha} + \frac{\eta^{\gamma+\beta}}{S}(1-\chi_0) + \overline{\eta}\right) + L_1^2 R^2(k\eta)C_2\overline{\eta}^{1+\gamma}\right)$$

Finally, by noting that $\eta^{2\alpha} + \frac{\eta^{\gamma+\beta}}{S}(1-\chi_0) + \overline{\eta} \geq \overline{\eta}^{1+\gamma}$ we obtain the desired result. $\qquad\square$

*Proof of Theorem 2.* Let's introduce constant $\delta = \eta^{2\alpha} + \frac{\eta^{\gamma+\beta}}{S}(1-\chi_0) + S\eta$ (appears first in Lemma 7; $\chi_0 = 1$ iff $\epsilon_k$ is Gaussian (see Assumption 1)). We want to produce a bound on $\mathcal{W}_2(\mathcal{L}(X_{k'}), \mathcal{L}(Z_{\eta k'}))$. To do so we consider $k' = Sk + i, i < S$ and rewrite it as:

$$\mathcal{W}_2(\mathcal{L}(X_{k'}), \mathcal{L}(Z_{k'\eta})) \leq \mathcal{W}_2(\mathcal{L}(X_{k'}), \mathcal{L}(X_{Sk})) + \mathcal{W}_2(\mathcal{L}(X_{Sk}), \mathcal{L}(Z_{\eta k'}))$$

We note that $L_2$ norm between random variables upper bounds Wasserstein-2 distance between their distributions by definition of the metric. Therefore, the first term is bounded by Lemma 6 as

$$\mathcal{W}_2(\mathcal{L}(X_{k'}), \mathcal{L}(X_{Sk})) \leq C'^{\frac{1}{2}}S\eta \leq C'^{\frac{1}{2}}\delta^{\frac{1}{2}}.$$

To bound the second one, we consider the following trick:

$$\mathcal{W}_2(\mathcal{L}(X_{Sk}), \mathcal{L}(Z_{\eta k'}))$$
$$\leq \mathcal{W}_2(\mathcal{L}(X_{Sk}), \mathcal{L}(Y_{\overline{\eta}k}^X)) + \mathcal{W}_2(\mathcal{L}(Y_{\overline{\eta}k}^X), \mathcal{L}(Y_{\eta k'}))$$
$$+ \mathcal{W}_2(\mathcal{L}(Y_{\eta k'}), \mathcal{L}(Z_{\eta k'}^Y)) + \mathcal{W}_2(\mathcal{L}(Z_{\eta k'}^Y), \mathcal{L}(Z_{\eta k'}))$$

The first term is bounded by Lemma 7:

$$\mathcal{W}_2(\mathcal{L}(X_{Sk}), \mathcal{L}(Y_{\overline{\eta}k}^X)) \leq C''R(k'\eta)\delta^{\frac{1}{2}},$$

where $C''$ is defined in Lemma 7). The second one can be bounded by Lemma 8 as

$$\mathcal{W}_2(\mathcal{L}(Y_{\overline{\eta}k}^X), \mathcal{L}(Y_{\eta k'})) \leq C_2(S\eta)^{\frac{1+\gamma}{2}}R(k'\eta) \leq C_2 R(k'\eta)\delta^{\frac{1}{2}}.$$

The third one by Lemma 9 and then by Lemma 7 by the same upper bound exactly as the second one. To bound the last one we use entropy bound on $\mathcal{W}_2$ (equation 6):

$$\mathcal{W}_2(\mathcal{L}(Z_{\eta k'}^Y), \mathcal{L}(Z_{\eta k'})) \leq \mathcal{C}_{\mathcal{W}}\left(\mathcal{L}(Z_t)\right)\left(D_{\mathrm{KL}}^{\frac{1}{2}}\left(\mathcal{L}(Z_{\eta k'}^Y)\middle\|\mathcal{L}(Z_{\eta k'})\right) + D_{\mathrm{KL}}^{\frac{1}{4}}\left(\mathcal{L}(Z_{\eta k'}^Y)\middle\|\mathcal{L}(Z_{\eta k'})\right)\right),$$

and by using Corollary 6 with Lemma 4 and Remark 2 we bound it as:

$$\mathcal{W}_2(\mathcal{L}(Z_{\eta k'}^Y), \mathcal{L}(Z_{\eta k'}))$$

$$\leq \eta^{\frac{\gamma}{2}} \left(\frac{2d}{M_0}\right)^{\frac{1}{2}} \sigma_1 e^{M_0 k' \eta} \left(C_3^{\frac{1}{2}} \eta^{-\frac{\gamma}{2}} R(k'\eta)(k'\eta)^{\frac{1}{2}} \delta^{\frac{1}{2}} + 2^{-\frac{1}{4}} C_3^{\frac{1}{4}} \eta^{-\frac{\gamma}{4}} R^{\frac{1}{2}}(k'\eta)(k'\eta)^{\frac{1}{4}} \delta^{\frac{1}{4}}\right)$$

$$\leq \left(\frac{2d}{M_0}\right)^{\frac{1}{2}} \sigma_1 e^{M_0 k' \eta} \left(C_3^{\frac{1}{2}} R(k'\eta)(k'\eta)^{\frac{1}{2}} \delta^{\frac{1}{2}} + 2^{-\frac{1}{4}} C_3^{\frac{1}{4}} R^{\frac{1}{2}}(k'\eta)(k'\eta)^{\frac{1}{4}} \eta^{\frac{\gamma}{4}} \delta^{\frac{1}{4}}\right)$$

Finally, we obtain the bound between $\mathcal{W}_2(\mathcal{L}(X_{k'}), \mathcal{L}(Z_{k'\eta}))$ by summing up those bounds and simplifying:

$$\mathcal{W}_2(\mathcal{L}(X_{k'}), \mathcal{L}(Z_{k'\eta}))$$
$$\leq \Big(\big(C_4(k'\eta)^{\frac{1}{2}} e^{M_0 k'\eta} + C_5\big) R(k\eta) + C'^{\frac{1}{2}}\Big) \delta^{\frac{1}{2}} + C_6(k'\eta)^{\frac{1}{2}} e^{M_0 k'\eta} R^{\frac{1}{2}}(k\eta) \eta^{\frac{\gamma}{4}} \delta^{\frac{1}{4}},$$

where $C_4 = \left(\frac{2d}{M_0}\right)^{\frac{1}{2}} \sigma_1 C_3^{\frac{1}{2}}$, $C_5 = C'' + 2C_2$, $C_6 = \left(\frac{2^{\frac{1}{2}} d}{M_0}\right)^{\frac{1}{2}} \sigma_1 C_3^{\frac{1}{4}}$ and $R(k'\eta) \leq e^{4(M+1)k'\eta} \sqrt{1 + \|x_0\|^2}$ by Lemma 5.

Recall that $\delta = \eta^{2\alpha} + \frac{\eta^{\gamma+\beta}}{S}(1 - \chi_0) + S\eta$, thus, to eliminate $S$ we set $S = \eta^{-\frac{1-\beta}{2}}(1 - \chi_0) + \chi_0$. Observe that $S\eta = \eta^{\frac{1+\beta}{2}}(1 - \chi_0) + \chi_0 \eta \leq 1$. Moreover, by defining $\theta = \min\left\{\alpha; \frac{(\gamma+1)(1+\chi_0)+(\gamma+\beta)(1-\chi_0)}{4}\right\}$, from Corollary 1 we have that $\delta \leq 3\eta^{2\theta}$. Substituting and re-arranging the constants yields the final expression of the form:

$$\mathcal{W}_2\big(\mathcal{L}(X_{k'}), \mathcal{L}(Z_{k'\eta})\big) = \mathcal{O}\Big(\big(1 + (k'\eta)^{\frac{1}{2}}\big) e^{\mathcal{O}(k'\eta)} \eta^\theta + (k'\eta)^{\frac{1}{4}} e^{\mathcal{O}(k'\eta)} \eta^{\frac{\theta}{2}+\frac{\gamma}{4}}\Big),$$

where constants depend only on ones defined in Assumption 1.

$\square$

