# OpenReview forum: "Ito Diffusion Approximation of Universal Ito Chains for Sampling, Optimization and Boosting"
_ICLR.cc/2024/Conference — ICLR 2024 poster_

### Official Review · Reviewer_4gNA · 2023-10-29

**Soundness:** 1 poor
**Presentation:** 2 fair
**Contribution:** 1 poor
**Rating:** 3
**Confidence:** 4

**Summary:**

This work considers a rather general and broad class of Markov chains,
Ito chains that look like Euler-Maryama discretization of some Stochastic
Differential Equation. However, I am not fully convinced of the importance of this work.

**Strengths:**

This paper studies a rather general and broad class of Markov chains and studies the convergence rates.

**Weaknesses:**

1. The theoretical results do not provide any new insight to the community.
2. The analysis is based on standard techniques.
3. The paper lacks a empirical study to verify the theoretical findings.

**Questions:**

None

---

> ### Author Response · Authors · 2023-11-18
>
> Thank you for review. Let us address the concerns that you raised:
>
> >The theoretical results do not provide any new insight to the community.
>
> Prior to our knowledge, our work is the first one to establish the following insights:
> 1. **Convergence in Wasserstein-2 of SGD diffusion** under the weak assumption that $\mathcal{W}_2$-CLT holds, which in particular, satisfied if the noise satisfies $\mathbb{E}\|\epsilon_k\|^4 \le \mathrm{const}$, which in prior works appeared only in (Li et al., 2019) [38] (last row in Table 2), however in that work convergence was proved in the weak sense and assumed dissipativity of the drift, which is a restriction, that does not allow to apply to such to problems as a stochastic approximation or saddle-point optimization.
> 2. **We are the first** to establish **non-asymptotic convergence for the Stochastic Gradient Langevin Boosting** algorithm, developed in [53]. All prior results around SGLD do not apply to that case, as they all assume the normality of the noise, which is not satisfied in that case. In the original work, authors were able to obtain only asymptotic first-order weak convergence results based on the almost a century-old work of Kushner J., "On the Weak Convergence of Interpolated Markov Chains to a Diffusion."
>
> >The analysis is based on standard techniques.
> 1. **All prior works on the convergence of Markov Chains to Diffusions assume at least distant dissipativity** of the drift (that drift points to the origin outside of a ball), which simplifies their analysis, and **most of the works assume normality of the noise** or close to normal distribution, **or establish convergence in a weaker than Wasserstein-2 sense**. **Techniques developed in our work**, namely **window coupling (see Eq. 4) and coupling via noise (see Eq. 3)**, **are novel and non-standard**. They both aim to **circumvent the lack of such assumptions on which the prior works were based. **
> 2. Regarding standard results like the Girsanov theorem, it does not apply to our non-Markov process, which appears in our analysis due to our novel coupling approaches that procured a non-Markov process for which the Novikov condition fails. Thus, in Theorem 1, we showed how such a non-markovian process can be reduced to the standard via a non-standard chain of arguments developed in Section B.3., based on Lemma 10.  **Again, this is not only a non-standard technique but provides novel fundamental insights about the Girsanov theorem.**
> 3. As for the SGD diffusion case, our Lemma 4 and techniques used to prove it are crucial for establishing Wasserstein-2 convergence of the SGD diffusion, which is another novel development of our work, which is game-changing results for showing SGD diffusion bounds for Wasserstein-2, without which the final bound of Theorem 2, would've been trivial and no convergence established. Which is, **again, new insight and new technique.**
>
> >The paper lacks a empirical study to verify the theoretical findings.
>
> Our work is purely theoretical and establishes explicit convergence bounds for many algorithms known for practicians to exhibit diffusion-like behavior, like SGLB, SGD, and any stochastic optimization. We merely unified them under one analysis done at the weakest possible assumptions. Such unification produced novel, previously unknown/sharper bounds of their convergence in Wasserstein-2 for algorithms like SGD and SGLB.
>
> We are confused about what type of empirical study the Reviewer wants to see, as the Wasserstein-2 distance is non-constructive by its definition (see Section A), while it is of undoubtful importance for the field.

---

### Official Review · Reviewer_RRnG · 2023-10-29

**Soundness:** 3 good
**Presentation:** 3 good
**Contribution:** 3 good
**Rating:** 6
**Confidence:** 3

**Summary:**

This paper studies a broad class of Markov chains called Ito chains, which allows for almost arbitrary isotropic and state-dependent noise instead of normal and state-independent one. The paper analyzes the coupling between Ito chains and its anagolous SDEs, and provides an upper bound for $\mathcal{W}_2$ distance between laws of the two. Their results cover many of the known estimates. In particular, some techniques such as Window Coupling and an augmented version of Girsanov's Theorem are of independent interest.

**Strengths:**

Originality: The originality is high. The authors study a quite general class of Markov chains and analyze the upper bound between SDE, which is also a popular research area.
Clarity: the clarity is good. The paper sequentially introduces several auxiliary SDEs/Markov chains to eventually bridge the gap between the processes of interest: $X_k$ and $Z_t$. The motivation of each proposed auxiliary processes are clearly stated and is easy to follow for readers.
Significance: The theoretical contribution is quite good for this area. The authors consider non-Gaussian and state-dependent chains which is not so deeply investigated in literature. The window coupling and "modified" Girsanov theorem may be of independent interest.

**Weaknesses:**

There are some mistakes and typos in proofs:
 - In Appendix, the equations in the middle of page.18 (the one after (10)), a noise matrix $\sigma(Y_{\bar{\eta}k}^S)$ before $\zeta_k^S(X_{Sk})$ is missing. The same issue appears in most of the later equations in this proof.
 - In the same equation, the last two terms have a wrong scaling w.r.t. $S$. Specifically, they should be $4\mathbb{E}\left[\left\|\frac{1}{\sqrt{S}}\sum_{i=1}^{S-1} \sigma(X_{Sk}) (\epsilon_{Sk+i} - \frac{1}{\sqrt{S}}\zeta_k^S(X_{Sk})) \right\|^2\right]$ and $4\mathbb{E}\left[\left\|
\sum_{i=0}^{S-1}(\sigma(X_{Sk}) - \sigma(Y_{\bar{\eta}k}^X)) \zeta_k^S(X_{Sk}) \right\|^2\right]$. Consequently, some following terms should be multiplied by $S^2$ or so. (It seems not to affect the final upper bound, though. It is better to still check the whole roadmap of proofs.)
 - I wonder how the last term in the top equation on page.19 is derived ($4M_0\|X_{Sk}-Y_{\bar{\eta}k}^X\|^2$). It seems merely using Assumption 1 (4) is not enough. Or maybe there should be some additional assumptions?

**Questions:**

The window coupling seems to be among the crucial parts in this paper. Could authors explain if this is first proposed by this paper, or already appears in literature? It is best to provide some references.

The paper gives better bounds for SGD with Gaussian/non-Gaussian cases [1]. Could authors compare the theoretical techniques used in two papers and explain where do such improvements probably come from?

---

> ### Author Response · Authors · 2023-11-18
>
> We are grateful to the Review for the positive review and fair concerns. Let us answer the weaknesses first.
> > In Appendix, the equations in the middle of page.18 (the one after (10)), a noise matrix .. is missing. ...
>
>
> >In the same equation, the last two terms have a wrong scaling w.r.t. $S$...
>
> Thank you for noticing that typos! As you mentioned, it does not affect the final bound as we are reducing noise to a form passed into the $\mathcal{W}_2$-CLT assumption. **We fixed both typos in the updated revision** by keeping $\zeta^{S}_k$ outside of summation, which also improved clarity.
>
> >I wonder how the last term in the top equation on page.19 is derived. It seems merely using Assumption 1 (4) is not enough. Or maybe there should be some additional assumptions?
>
> Assumption 1 (4) and (5) are enough! Seems like during pre-submission revisioning we lost a chain of arguments, where we show how it follows. **We updated the revision** with the missing argument. The argument is following:
> First we note the following algebraic identities:
>
>  $\mathbb{E}\langle \sigma(x) (\epsilon_{k}(x) - \epsilon_{k}(x')), \sigma(x) (\epsilon_{k}(x) + \epsilon_{k}(x'))\rangle = \mathbb{E}\langle \sigma(x') (\epsilon_{k}(x) - \epsilon_{k}(x')), \sigma(x') (\epsilon_{k}(x) + \epsilon_{k}(x'))\rangle = 0$
>
> $\mathbb{E} \big\| (\sigma(x) + \sigma(x'))(\epsilon_{k}(x) - \epsilon_{k}(x'))\|^2 \ge \mathbb{E} \big\| (\sigma(x) - \sigma(x'))(\epsilon_{k}(x) - \epsilon_{k}(x'))\|^2$
>
> They hold due to assumptions of $\mathbb{E}\epsilon_k = 0$, $\mathbb{E} \epsilon_k \epsilon_k^T = I_{d}$ and $\sigma = \sigma^T \ge 0$. Then it follows:
> $$M_{0}^2 \|x-x'\|^2 \ge \mathbb{E}\|\sigma(x) \epsilon_{k}(x) - \sigma(x') \epsilon_{k}(x')\|^2 $$
>   $$  =   \frac{1}{4} \mathbb{E}\Big\| \big(\sigma(x) + \sigma(x'))(\epsilon_{k}(x) - \epsilon_{k}(x')) + \big(\sigma(x) - \sigma(x'))(\epsilon_{k}(x) + \epsilon_{k}(x'))\Big\|^2$$
>    $$ =   \frac{1}{4} \mathbb{E}\Big\| \big(\sigma(x) + \sigma(x'))(\epsilon_{k}(x) - \epsilon_{k}(x'))\Big\|^2 + \frac{1}{4}\mathbb{E}\|\big(\sigma(x) - \sigma(x'))(\epsilon_{k}(x) + \epsilon_{k}(x'))\Big\|^2 $$
>     $$ + \frac{1}{2}\mathbb{E}\Big\langle \big(\sigma^2(x)
>     - \sigma^2(x')\big) (\epsilon_{k}(x) - \epsilon_{k}(x')), \epsilon_{k}(x) - \epsilon_{k}(x')\Big\rangle $$
>    $$ =  \frac{1}{4} \mathbb{E}\Big\| \big(\sigma(x) + \sigma(x')\big)(\epsilon_{k}(x) - \epsilon_{k}(x'))\Big\|^2 + \frac{1}{4}\mathbb{E}\|\big(\sigma(x) - \sigma(x'))(\epsilon_{k}(x) + \epsilon_{k}(x'))\Big\|^2 $$
>     $$ + \frac{1}{2}\mathbb{E}\Big\langle \sigma(x) (\epsilon_{k}(x) - \epsilon_{k}(x')), \sigma(x)(\epsilon_{k}(x) - \epsilon_{k}(x')\Big\rangle $$
>     $$ - \frac{1}{2}\mathbb{E}\Big\langle \sigma(x') (\epsilon_{k}(x) - \epsilon_{k}(x')), \sigma(x)(\epsilon_{k}(x) - \epsilon_{k}(x')\Big\rangle$$
>    $$ =  \frac{1}{4} \mathbb{E}\Big\| \big(\sigma(x) + \sigma(x')\big)(\epsilon_{k}(x) - \epsilon_{k}(x'))\Big\|^2 + \frac{1}{4}\mathbb{E}\|\big(\sigma(x) - \sigma(x'))(\epsilon_{k}(x) + \epsilon_{k}(x'))\Big\|^2 $$
>   $$  \ge  \frac{1}{4} \mathbb{E}\Big\| \big(\sigma(x) - \sigma(x')\big)(\epsilon_{k}(x) - \epsilon_{k}(x'))\Big\|^2 + \frac{1}{4}\mathbb{E}\|\big(\sigma(x) - \sigma(x'))(\epsilon_{k}(x) + \epsilon_{k}(x'))\Big\|^2 $$
>   $$  \ge  \frac{1}{2} \mathbb{E} \big\|(\sigma(x) - \sigma(x'))\epsilon_{k}(x)\big\|^2 + \frac{1}{2} \mathbb{E} \big\|(\sigma(x) - \sigma(x'))\epsilon_{k}(x')\big\|^2 $$
>     $$\ge \|\sigma(x) - \sigma(x')\|_{F}^2$$
>
> And on questions:
> > The paper gives better bounds for SGD with Gaussian/non-Gaussian cases [1]. Could authors compare the theoretical techniques used in two papers and explain where do such improvements probably come from?
>
> If we understood correctly, by [1], you mean Alfonsi A. et al., "Optimal transport bounds between the time-marginals of a multidimensional diffusion and its Euler scheme.".
> To answer this, we can refer to Remark 3.2 in [1], where the authors point out that the bound can be improved if uniform ellipticity is assumed (which we explicitly stated in Table 2 and our Assumption 1). We also note that the work [1] considered a general SDE case and might not cover the SGD case. As for the SGD case, we compare it against [15] and note that the work [15] considers the Wasserstein-1 distance. We believe that improvement in our bound comes mainly from using entropic bounds instead of the Lyapunov-type of analysis, which became possible due to Lemma 4.
>
> >The window coupling seems to be among the crucial parts in this paper. Could authors explain if this is first proposed by this paper, or already appears in literature? It is best to provide some references.
>
> Best to our knowledge, we have not seen it in any prior works, however, some intuitive kind of arguments similar to that are common in prior literature (e.g., Latz J., "Analysis of stochastic gradient descent in continuous time"). As for the term $L(X_{Sk} - Y_k)$, since all prior works assumed dissipativity/convexity, it was not needed for them.

---

### Official Review · Reviewer_i2Nt · 2023-11-01

**Soundness:** 3 good
**Presentation:** 3 good
**Contribution:** 3 good
**Rating:** 8
**Confidence:** 3

**Summary:**

This paper considers a general class of Markov chains called Ito chains that cover a wide range of applications including sampling, optimization, and boosting. The paper provides bounds for the approximation error in W_2 distance between such Ito chains and the corresponding stochastic differential equation (which can then be used to study the Markov chain). In several applications, the bounds improve upon previous results or are completely new.

**Strengths:**

The results in this paper provide novel and general bounds on the approximation error for Ito chains using the corresponding stochastic differential equation. The results cover a broad range of applications in sampling, optimization, and boosting (for example, SGLD, SGD, and Stochastic Gradient Boosting) and are novel in several such applications (for example, the results cover almost arbitrary isotropic and state-dependent noise). The proof involves several new ideas and the presentation is quite clear.

**Weaknesses:**

It would be nice if the author(s) could comment on whether Assumption 1 in Section 2 is typically satisfied/easy to verify in applications.

**Questions:**

It would be nice if the author(s) could comment on whether Assumption 1 in Section 2 is typically satisfied/easy to verify in applications.

---

> ### Author Response · Authors · 2023-11-18
>
> We are grateful for your positive review and high score!
>
> Let us answer your question/weakness.
>
> >It would be nice if the author(s) could comment on whether Assumption 1 in Section 2 is typically satisfied/easy to verify in applications.
>
> When formulating assumptions, we attempted to cover as much generality as possible to cover/generalize the majority of prior works.
>
> As for checking them, it is helpful to simplify some of them first (with maybe loss of generality), for example:
> 1. $\mathcal{W}_{2}$-CLT will be satisfied if $\mathbb{E} \epsilon_k^4 \le \mathrm{const}$
> 2. For uniform ellipticity, if we are working with SGD with dissipative/convex loss (e.g., regularized by $L_{2}$ norm), then it is possible to show that iterations remain in a ball, which, even if the covariance is not bounded by default, can be considered as bounded without loss of generality, which is a standard trick in SGD literature.
> 3. Assumption like $\mathbb{E} \| \sigma(x) \epsilon_k(x) - \sigma(x') \epsilon_k(x')\|^2 \le M_{0}^2 \|x-x'\|^2$ will be satisfied if both $\sigma(x), \epsilon_k(x)$ are Lipshitz-continuous, which in the case of SGD/SGLD is satisfied automatically if the loss (which gradient we are taking as drift $b(x)$ are Lipshitz smooth)
> 4. Assumptions on bias/covariance shifts in such cases as smoothed SGDL or SGD will be satisfied automatically if the loss function (which gradient we are taking as drift $b(x)$) is Lipshitz-smooth, which is quite a standard assumption.
> 5. Generally, checking the assumptions will require proving they are held on a case-by-case basis. However, in most cases, there is something like Lipshitz-smoothness of the loss / Lipshitz continuity of the drift, which makes them hold by default.

---

> > ### Comment · Reviewer_i2Nt · 2023-11-22
> >
> > Thank you for the detailed response and clarification!

---

### Meta-Review · Area_Chair_8vCN · 2023-12-14

**Metareview:**

Authors study chains that look like Euler-Maryama discretization of Ito diffusions. The method can handle a wide range of diffusions with state-dependent noise as well as inexact drift and diffusion coefficient, which does include SGLD etc.
The papers main contribution is a theoretical control over the distance between the chain and the underlying diffusion. Although similar results already exist in the literature, their analysis has novelty in certain settings.

This paper was reviewed by 3 reviewers and received the following Rating/Confidence scores: 8/3, 6/3, 3/4. The reviewer who is championing the paper has a rather low confidence score and the one strongly rejecting the paper has valid concerns. However, they did not participate in a discussion, as such AC is down-weighting the score. The authors should clearly highlight the novelty in their analysis, e.g. by improving the table in the paper. In addition, authors are missing recent literature on sampling with Langevin Monte Carlo.

I think the paper is overall interesting and should be included in ICLR. The authors should carefully go over and clarify all reviewers' questions/concerns.

**Justification For Why Not Higher Score:**

I think the paper is borderline.

**Justification For Why Not Lower Score:**

n/a

---

### Decision · Program_Chairs · 2024-01-16

Accept (poster)